# Maternal diet-induced obesity during pregnancy alters lipid supply to mouse E18.5 fetuses and changes the cardiac tissue lipidome in a sex-dependent manner

Lucas C Pantaleão[1]*[†], Isabella Inzani[1]*[†], Samuel Furse[1,2], Elena Loche[1], Antonia Hufnagel[1], Thomas Ashmore[1], Heather L Blackmore[1], Benjamin Jenkins[1,2], Asha A M Carpenter[1], Ania Wilczynska[3,4], Martin Bushell[3,4], Albert Koulman[1,2], Denise S Fernandez-Twinn[1], Susan E Ozanne[1]*

[1]Wellcome-MRC Institute of Metabolic Science and Medical Research Council Metabolic Diseases Unit, University of Cambridge, Addenbrooke's Hospital, Cambridge, United Kingdom; [2]Core Metabolomics and Lipidomics Laboratory, Wellcome-MRC Institute of Metabolic Science, University of Cambridge, Addenbrooke's Treatment Centre, Cambridge, United Kingdom; [3]Cancer Research UK Beatson Institute, Glasgow, United Kingdom; [4]Institute of Cancer Sciences, University of Glasgow, Glasgow, United Kingdom

*For correspondence:
lp435@medschl.cam.ac.uk (LCP);
ii233@medschl.cam.ac.uk (II);
seo10@cam.ac.uk (SEO)

[†]These authors contributed equally to this work

Competing interest: The authors declare that no competing interests exist.

**Abstract** Maternal obesity during pregnancy has immediate and long-term detrimental effects on the offspring heart. In this study, we characterized the cardiac and circulatory lipid profiles in late gestation E18.5 fetuses of diet-induced obese pregnant mice and established the changes in lipid abundance and fetal cardiac transcriptomics. We used untargeted and targeted lipidomics and transcriptomics to define changes in the serum and cardiac lipid composition and fatty acid metabolism in male and female fetuses. From these analyses we observed: (1) maternal obesity affects the maternal and fetal serum lipidome distinctly; (2) female fetal heart lipidomes are more sensitive to maternal obesity than males; (3) changes in lipid supply might contribute to early expression of lipolytic genes in mouse hearts exposed to maternal obesity. These results highlight the existence of sexually dimorphic responses of the fetal heart to the same in utero obesogenic environment and identify lipids species that might mediate programming of cardio-vascular health.

## Editor's evaluation

The manuscript by Pantaleao et al., describes the effects of maternal diet-induced obesity on lipid composition in maternal and fetal serum and the fetal heart, and in the fetal heart transcriptome. The results presented provide insight into the still poorly understood processes influencing the long-term health of the fetus. A limitation is that this is a largely descriptive study. Nonetheless, the authors provide a detailed description of the lipid composition changes in response to maternal obesity and associated with sex.

## Introduction

Mammalian heart development and maturation involve a complex array of processes that are only completed postnatally, when increased systemic demands and changes in substrate and oxygen availability promote major cardiac remodelling (Reviewed by *Piquereau and Ventura-Clapier, 2018*). Appropriate and regulated flow of hormones, nutrients, metabolites and absorbed gases into fetal tissues is required to ensure that intrauterine development is achieved in an appropriate time-sensitive manner. Therefore, adverse gestational conditions – such as maternal obesity – can disrupt maternal/fetal molecule interchange, leading to impaired fetal development, which can have long-term impacts on cardio-metabolic health postnatally (*Dong et al., 2013*; *Zambrano and Nathanielsz, 2013*). Such a causal link between maternal metabolic status and lifelong offspring health and disease is encompassed in what has been termed the Developmental Origins of Health and Disease (*Barker, 2007*).

Maternal obesity during gestation is one condition that has been shown to raise the risk of noncommunicable diseases in the expectant mother and her children. Numerous studies in humans and animal models suggest that obesity during pregnancy has immediate and long-term detrimental effects including increased risk of congenital heart disease (*Helle and Priest, 2020*), and increased susceptibility of the offspring to cardiometabolic abnormalities postnatally (*Guénard et al., 2013*; *Loche et al., 2018*). This is of particular importance, as recent data indicates that around 50% of pregnant women in developed countries are currently either overweight or obese, and cardiovascular diseases are a leading cause of death worldwide (*NMPA Project Team, 2019*; *GBD 2017 Causes of Death Collaborators, 2018*).

Mild hypoxia, high maternal insulin and leptin levels, as well as changes in nutrient and metabolite availability have been implicated as causal factors, mediating the effects of maternal obesity on the fetus and its long-term health (*Howell and Powell, 2017*). However, there is limited data in relation to the molecular consequences of such exposure on the fetal heart. Although a small number of studies have provided some evidence that maternal obesity affects the fetal cardiac transcriptome and protein profile, these studies do not explain the whole complexity of changes in the cardiac phenotype. In addition, there is very limited data on the contribution of the maternal and fetal lipidome to programming mechanisms (*Catalano and Shankar, 2017*).

Lipids are a complex group of structural, energy and signaling molecules involved in a variety of physiological, metabolic, and pathological processes. Changes in the murine heart lipidome have been shown to initiate and promote inflammatory reactions after infarction (*Halade et al., 2018*), and cellular lipid composition has been associated with the distinction between physiological and pathological cardiac hypertrophy and with prognosis of cardiac disease (*Tham et al., 2018*; *Le et al., 2014*). Moreover, recent studies explored the association of the cardiac lipidome with life stage progression, showing how changes in intracellular lipids contribute to heart maturation at birth (*Walejko et al., 2018*) and the impact of ageing on the heart (*Eum et al., 2020*).

Despite the growing interest in lipid profiling in health and disease, the study of fetal cardiac lipids in the context of maternal obesity models remains largely unexplored. The aim of the current study was therefore to use lipidomics and cardiac transcriptomics to identify lipid pathways that may be associated with developmental phenotypes that lead to chronic diseases later in life. In addition, given the growing evidence for sex-specific differences in the programming field, a further aim was to establish if any of the responses were sexually dimorphic. As circulating lipids are often used as biomarkers of cardiovascular health, we also sought to investigate associations between maternal and fetal serum lipidomes.

## Results

### Maternal obesity affects mouse offspring heart morphology

A well-established diet-induced maternal obesity model in which the female mouse develops obesity and gestational diabetes was used to investigate the effects of maternal obesity on the offspring heart. At gestational day 18.5, male and female fetuses from obese dams were smaller than controls (*Figure 1A*), and although their heart weights were not significantly different from control hearts in either absolute or relative terms (*Figure 1B and C*), obese fetuses showed smaller left, but not right, ventricular wall volume (*Figure 1D, E and F*). The change in left ventricular wall volume was not significant when expressed as relative to fetal body mass, indicating symmetric growth restriction in the

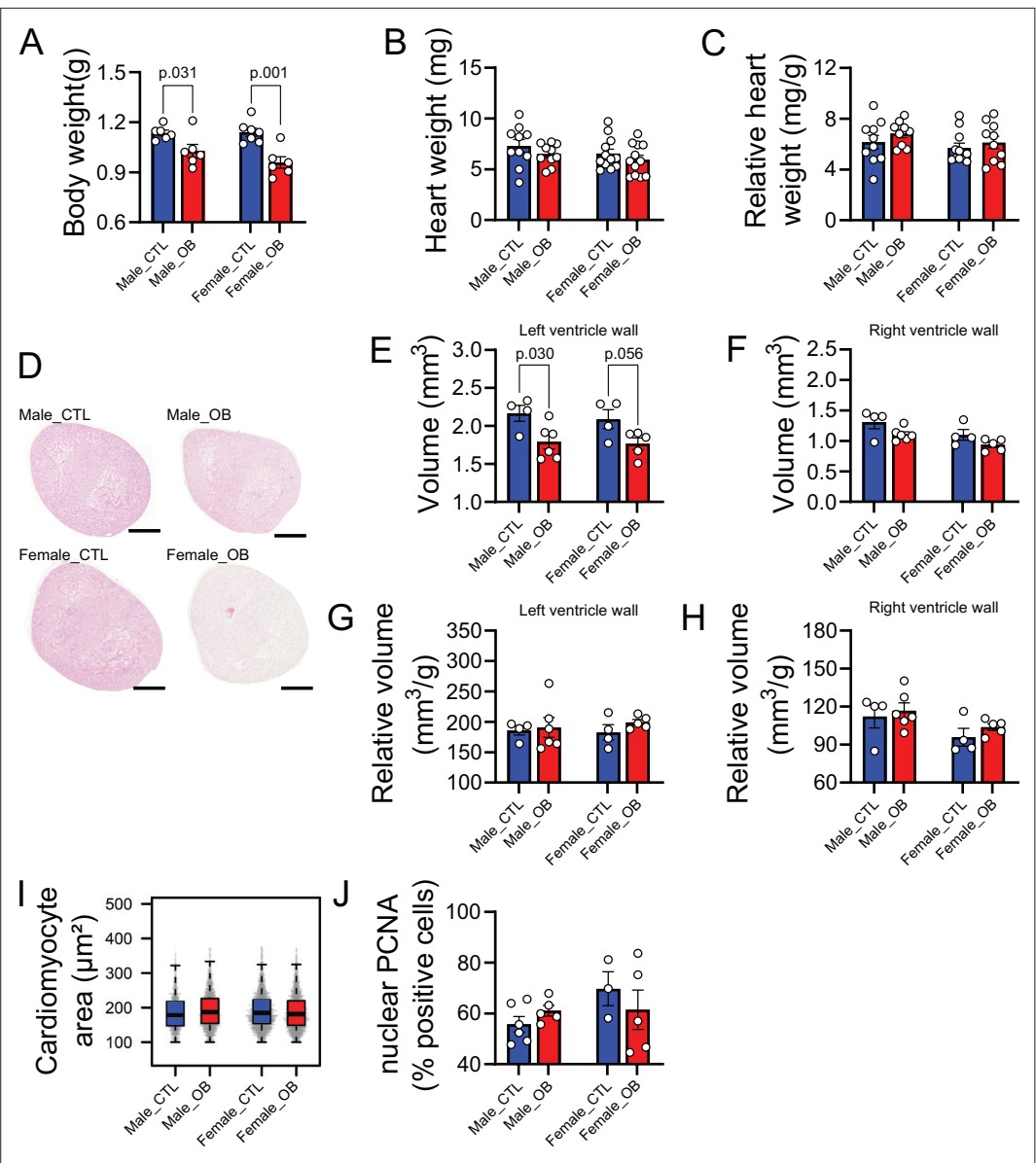

**Figure 1.** Fetal characteristics at gestational day 18.5. (**A**) Body weight of male and female fetuses from healthy control (CTL) and obese (OB) mouse dams at gestational day 18.5. Male CTL n = 6, male OB n = 6, female CTL n = 7, female OB n = 6. (**B–C**) Heart weight and heart weight/litter average body weight ratio of male and female fetuses from CTL and OB dams at gestational day 18.5. Male CTL n = 10, male OB n = 10, female CTL n = 12, female OB n = 12. (**D**) Histological sections stained with eosin of male and female fetuses from CTL and OB dams at gestational day 18.5 used in cardiac stereology (see also *Figure 1—figure supplement 1*). Scale bar indicates 500 μm. (**E–H**) Left and right ventricular wall volume and ventricular wall volume/body weight ratio of male and female fetuses from healthy control (CTL) and obese (OB) mouse dams at gestational day 18.5. Male CTL n = 4, male OB n = 6, female CTL n = 4, female OB n = 5. (**I**) Boxplot showing median and quartiles of cardiomyocyte area distribution in male and female fetuses from CTL and OB dams at gestational day 18.5. Under the boxplot a beeswarm plot shows individual area of each cardiomyocyte analysed. Male CTL n = 6, male OB n = 7, female CTL n = 6, female OB n = 6 (see *Figure 1—figure supplement 2* for a representative image of stained cardiomyocytes). (**J**) Nuclear PCNA-positive cells percentage in hearts of male and female fetuses from CTL and OB dams at gestational day 18.5. Male CTL n = 6, male OB n = 5, female CTL n = 3, female OB n = 5. In panels A-H and J, p-values were calculated by Student t-test. In panel I, p-values were calculated using linear mixed-effects model followed by Tukey's *post-hoc* test.

The online version of this article includes the following source data and figure supplement(s) for figure 1:

*Figure 1 continued on next page*

*Figure 1 continued*

**Source data 1.** Sex-specific fetal body weight at E18.5 (*Figure 1A*).

**Source data 2.** Fetal heart weight at E18.5 (*Figure 1B*).

**Source data 3.** Fetal heart weight at E18.5 relative to the average litter weight (*Figure 1C*).

**Source data 4.** Left ventricle wall volume (*Figure 1E*).

**Source data 5.** Right ventricle wall volume (*Figure 1F*).

**Source data 6.** Relative left ventricle wall volume (*Figure 1G*).

**Source data 7.** Relative right ventricle wall volume (*Figure 1H*).

**Source data 8.** Fetal cardiomyocyte area (*Figure 1I*).

**Source data 9.** % Nuclear PCNA-positive cardiac cells (*Figure 1J*).

**Figure supplement 1.** Stereological analysis of the gestational day 18.5 fetal heart.

**Figure supplement 2.** Representative image of immunofluorescence identifying cardiomyocytes using a mAb anti-Cardiac Troponin I IgG followed by incubation with an Alexa Fluor 488-conjugated secondary IgG.

fetuses from obese pregnancies (*Figure 1G and H*). The observed reduction in ventricular size was not accompanied by changes in cardiomyocyte cell size (*Figure 1I*) or proliferation (*Figure 1J*) at this stage of development.

## Maternal obesity drives changes to the lipid composition of maternal and fetal serum

Maternal and fetal serum lipidomes were obtained by direct infusion high-resolution mass spectrometry, a rapid method used to profile the lipids in an organic extract. Full data and annotation of isobaric signals are available in the supplementary information (*Supplementary file 1*, *Figure 2— source data 3*). We used Principal Component Analysis (PCA) to identify orthogonal distance and relatedness amongst individual fetal and maternal serum lipidomes. This multivariate analysis demonstrated that there was a clear distinction between maternal and fetal serum lipid profiles, regardless of the maternal nutritional status or offspring sex (*Figure 2—figure supplement 1A*). In order to test the hypothesis that the lipid composition in the serum of the dams and her fetuses differed between control and obese mothers, PCAs of just these pairs of groups were performed. These suggested that there was clear segregation in each case driven by maternal dietary status (*Figure 2—figure supplement 1B and C*).

In order to identify the lipid pathways altered, we summed the abundance of the lipid variables in each lipid class (head group, assuming even chain length for the fatty acid, see *Supplementary files 1 and 2* for lipid signals used for each class) and calculated which classes differed in abundance according to maternal status. This showed that cholesteryl esters (CE), ceramides (Cer), and sphingomyelins (SM) were more abundant in serum from obese dams than in serum from controls (*Figure 2A*). Amongst fetal serum, changes to the abundance of lipid classes were generally similar between males and females. Serum levels of phosphatidylcholines/phosphatidylethanolamines and ceramides were significantly reduced, and phosphatidylglycerols, phosphatidic acids, triglycerides and the ratio of triglycerides to phospholipids were significantly increased in response to maternal obesity (*Figure 2B*). We then used factorial analysis to establish any sex differences. This revealed sex differences in cholesteryl esters (*Figure 2C*), with males showing higher relative abundance compared to females (*Figure 2—figure supplement 2A*).

## Specific lipid profiles differ between maternal and fetal serum

Lipid classes consist of several lipid isoforms that comprise different fatty acid residues. To assess the differences in lipids at the individual level, we used both univariate and multivariate models to identify differences between serum from obese and control dams and their fetuses. Although ceramides were more abundant at the class level in obese dam serum, no individual ceramide isoform was statistically significantly different between groups (*Figure 2D and E*). At the individual isoform level, campesterol (ST 28:1;O) and many unsaturated triglycerides were less abundant in sera of obese dams. Similar to changes observed at the class level, the abundance of two isoforms of sphingomyelin and two individual cholesteryl esters was also increased in the sera of obese mothers. Some phosphatidylcholines/

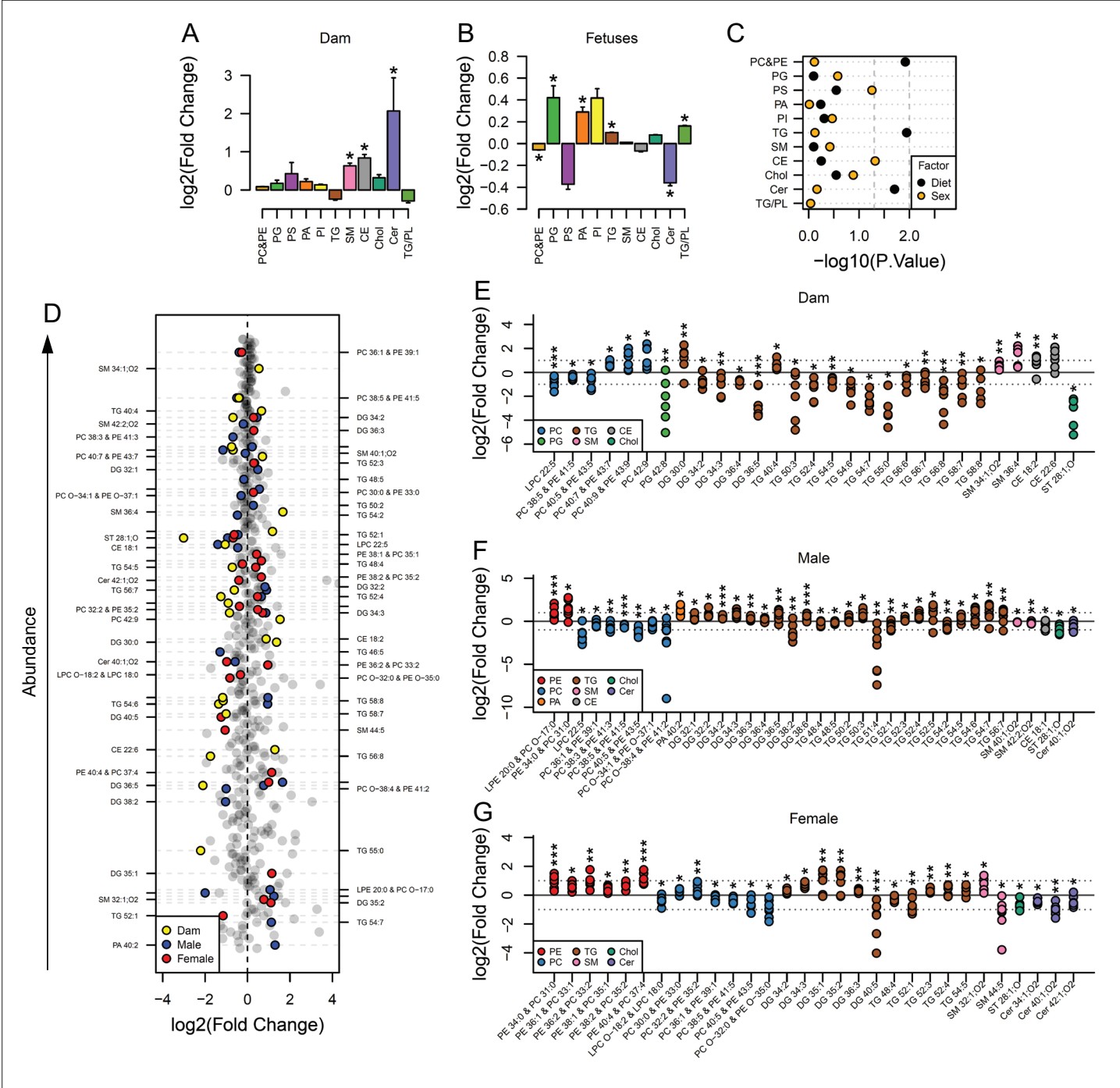

**Figure 2.** Maternal and fetal serum lipidome measured by direct infusion mass spectrometry. (**A–B**) Relative changes in serum lipid classes abundance in obese dams (**A**) and in obese fetuses (**B**). Values are mean + SE. *p < 0.05 calculated by Student t-test or Mann-Whitney test. (**C**) Influence of maternal diet and sex on fetal serum lipid classes abundance as calculated by factorial ANOVA. (**D**) Regulation of maternal and fetal serum lipid species ranked according to their abundance. Coloured dots represent statistically regulated species as calculated by univariate Student t-test (p < 0.05) and PLS-DA VIP (vip score >1) in maternal or fetal OB serum compared to CTL. (**E–G**) Serum levels of regulated lipids from obese dams (**E**) and from male (**F**) and female (**G**) fetuses of obese dams at gestational day 18.5. Each dot represents a result from one obese fetus' serum relative to the average of results for individual lipids in the control group (straight line). Dam CTL n = 9, dam OB n = 6, male fetuses CTL n = 10, male fetuses OB n = 8, female fetuses CTL n = 10, female fetuses OB n = 7; * p < 0.05, ** p < 0.01, *** p < 0.001 calculated by Student t-test. In figures A-C: PE, phosphatidylethanolamines/ odd chain phosphatidylcholines; PC, phosphatidylcholines/odd-chain phosphatidylethanolamines; PG, phosphatidylglycerols; PS, phosphatidylserines; PA, phosphatidic acids; PI, phosphatidylinositols; TG, monoglycerides, diglycerides and triglycerides; SM, sphingomyelins; CE, cholesteryl esters; Cer, ceramides; PL, phospholipids. In figures D-G, other isobaric lipids can contribute to these signals (*Supplementary file 1*). See also *Figure 2—figure*

*Figure 2 continued on next page*

*Figure 2 continued*

**supplement 1** and *Figure 2—figure supplement 2*.

The online version of this article includes the following source data and figure supplement(s) for figure 2:

**Source data 1.** Relative lipid classes abundance in maternal serum (*Figure 2A*).

**Source data 2.** Relative lipid classes abundance in fetal serum (*Figure 2B*).

**Source data 3.** Direct infusion high-resolution mass spectrometry of the serum (positive mode only) (*Figure 2D–G*).

**Figure supplement 1.** Multivariate analysis of the maternal and fetal serum lipid profiles.

**Figure supplement 2.** Serum levels of lipid groups.

odd chain phosphatidylethanolamines were more abundant, although two – PC 40:5/PE 43:5 and PC 38:5/PE 41:5 – and *lyso*-phosphatidylcholine (LPC) 22:5 were less abundant in obese dam sera.

We then assessed the correlation between maternal and fetal lipid composition and abundance using linear regression. We showed that only a few maternal phospholipids and *lyso*-phospholipids species – comprising LPC 18:2, 20:4, 22:5 and 22:6 – and campesterol were significantly correlated with the same species in both male and female fetal serum (*Figure 2—figure supplement 2B and C*). Contrasting to the signature observed in dam serum, naturally highly abundant triglyceride isoforms had increased abundance in both male and female fetuses with only a handful of exceptions (e.g. TG 54:2 and TG 48:5 in males – *Figure 2F*). Phosphatidylethanolamine/odd chain phosphatidylcholines isoforms were also more abundant in serum of both male and female fetuses of obese dams, with females showing increased abundance of the greatest number of species, and PE 34:0/PC 31:0 being regulated in a sex-independent manner (*Figure 2D, F and G*). Odd chain fatty acid containing phosphatidylcholines are isobaric with phosphatidylethanolamines (see *Supplementary file 2*); however, no other evidence from other lipid classes suggested a change in odd chain fatty acid metabolism. Consistent with the dam serum data, several phosphatidylcholine/odd chain phosphatidylethanolamines isoforms were less abundant in serum of male and female fetuses of obese dams.

## Maternal obesity is associated with changes in the fatty acid composition of phospholipids in maternal and fetal serum

The bulk of serum lipid species detected were glycerides, phospholipids and cholesterol (*Figure 2—figure supplement 2D*), with glycerides, phospholipids and *lyso*-phospholipids comprising three and two fatty acids respectively, that are covalently bonded to glycerol. As we observed differences in the abundance of individual lipid isoforms in maternal and fetal serum, we sought to elucidate whether the changes observed in fetuses from obese dams would also translate into an imbalance in the distribution of fatty acid residues in phospholipids. When clustered according to the number of double bonds as either saturated, monounsaturated or polyunsaturated residues, we observed that although the relative abundances of these were not significantly affected by maternal diet in fetal serum, feeding of the obesogenic diet during pregnancy was associated with increased relative monounsaturated fatty acids abundances in the maternal serum (*Figure 3A*), consistent with increased oleate observed in the dams (*Figure 3B*). We also made the observation that fatty acids from phospholipids with a chain length shorter than 18 carbons were generally more abundant in obese dam and fetal serum, whereas the abundance of longer molecules tended to be lower (*Figure 3B*).

At the individual level, we again identified a distinction between maternal and fetal serum profiles, with phospholipid fatty acid residues being differentially regulated (*Figure 3B*, see also *Figure 3—figure supplement 1*).

In line with the lower polyunsaturated phospholipid levels, those containing n-3 docosapentaenoic acid (DPA) (22:5) were less abundant, and those with saturated fatty acids myristic (14:0) and margaric (17:0), and monounsaturated myristoleic acid (14:1) were more abundant in the fetal sera of both sexes in response to maternal obesity. In contrast, oleic acid (18:1) from phospholipids was more abundant in the serum of obese dams only. Despite the differences observed, the fetal levels of a few residues were significantly correlated with the maternal ones, and an overall trend for positive correlation between fatty acids levels in maternal and fetal sera was observed (*Figure 3C*). By generating correlation matrices with group independent variables using Euclidian clustering, we also observed that several saturated and monounsaturated fatty acid residues were positively correlated and tended to cluster together (*Figure 3—figure supplement 2*). Similarly, polyunsaturated fatty

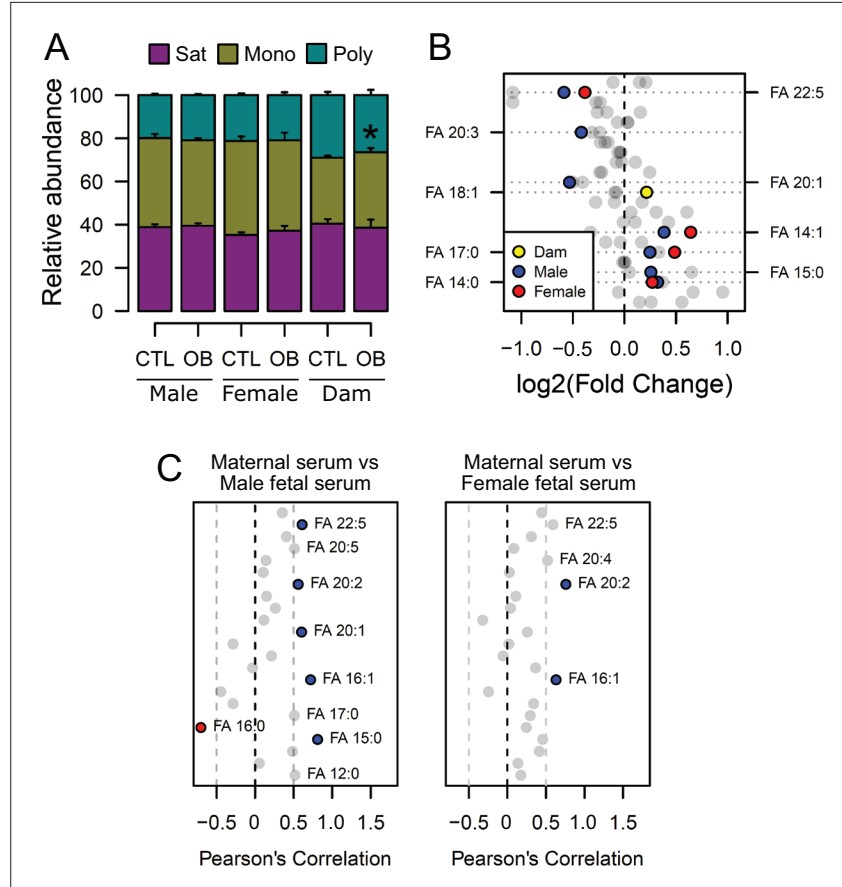

**Figure 3.** Fatty acid composition of serum phospholipids measured by direct infusion mass spectrometry using in-source CID fragmentation. (**A**) Grouped saturated, monounsaturated and polyunsaturated fatty acids content in maternal, male and female fetal serum at gestational day 18.5. Values are mean + SE. (**B**) Regulation of maternal and fetal serum fatty acids. Coloured dots represent statistically regulated fatty acids as calculated by univariate Student t-test or Mann-Whitney test (p < 0.05) in maternal or fetal OB serum compared to CTL. (**C**) Pearson's correlation between maternal serum fatty acids and the same fatty acids detected in the fetal serum. Blue and red dots represent species with significant positive and negative association (p < 0.05). Dam CTL n = 8, dam OB n = 6, male fetuses CTL n = 10, male fetuses OB n = 8, female fetuses CTL n = 8, female fetuses OB n = 6. See also *Figure 3—figure supplement 1* and *Figure 3—figure supplement 2*.

The online version of this article includes the following source data and figure supplement(s) for figure 3:

**Source data 1.** Relative fatty acid classes abundance in maternal and fetal serum (*Figure 3A*).

**Source data 2.** Fatty acids abundance obtained by direct infusion high-resolution mass spectrometry of the serum (negative mode) (*Figure 3B*).

**Figure supplement 1.** Radar plots showing regulation of fatty acids statistically changed in the serum or in the heart of fetuses from obese dams in different compartments.

**Figure supplement 2.** Correlation matrices showing Pearson's correlation between cardiac fatty acids in dams and fetuses.

---

acids established clusters and were positively correlated. Several saturated and monounsaturated fatty acids were negatively correlated to polyunsaturated species.

## Maternal obesity sex-specifically affects the fetal heart lipidome

We next sought to determine whether the lipid composition of fetal hearts was influenced by maternal obesity and whether its signature would follow a similar pattern to that observed in the fetal serum. Full cardiac lipidome data and annotation of isobaric signals are available in the supplementary information (*Supplementary file 2*, *Figure 4—source data 2* and *Figure 4—source data 4*). We observed that total cholesteryl esters were less abundant in both male and female fetal hearts from obese

pregnancies (*Figure 4A and B*). Total sphingomyelins were more abundant in both males and females, but the difference was only statistically significant in female hearts (*Figure 4B*). The observation that fewer lipid classes were perturbed in males than in females in response to maternal obesity was reproduced at the individual species level. Through sex specific PCAs, we observed a weaker degree of separation between control and obese individual lipidomes in males (*Figure 4C*) when compared to females (*Figure 4D*). In contrast, female heart lipidomes showed clear distinction between control and obese hearts, with more lipid species being significantly different between groups (42 compared to 18 in males) (*Figure 4E, F and G*).

Looking at the most abundant cardiac lipids, we observed that female hearts were more sensitive to change as a consequence of maternal obesity, with most lipids having greater fold change compared to male hearts (*Figure 4H*, see also *Figure 3—figure supplement 1* and *Figure 4—figure supplement 1*).

Lipid ontology analysis of the fetal heart lipidome using LION (*Molenaar et al., 2019*) also identified more biological features significantly enriched in female hearts compared to males (*Figure 4—figure supplement 2*). This analysis also revealed an overall decrease in phospholipids and *lyso*-lipids and, although not quantitatively significant, an increase in triglycerides in both males and females. Consistently, most modelled triglycerides were more abundant, and most phosphatidylcholines/odd chain phosphatidylethanolamines were less abundant in both male and female hearts exposed to in utero obesity (*Figure 4F and G*). Phosphatidylethanolamines/odd chain phosphatidylcholines were also regulated in both male and female hearts, and sphingomyelins were regulated in females only. Regarding the fatty acid composition of phospholipids, we did not observe changes in the overall content of fatty acid groups (*Figure 4I*). However, at the individual level, we observed lower abundance of DPA in both male and female hearts exposed to in utero obesity (*Figure 4J*). Male offspring also exhibited lower cardiac levels of $m/z$ 423.421, tentatively identified as the very long-chain saturated octacosanoic acid (28:0), although other metabolites can contribute to this signal. In females, we observed increased abundance of the highly abundant docosahexaenoic acid, FA 22:6, and lower abundance of palmitoleic, FA 16:1, and fatty acids FA 22:3 and FA 20:3.

## Maternal obesity induces changes in the fetal heart transcriptome to promote lipid metabolism

Changes in the abundance of individual lipid isoforms and the fatty acid composition of lipid classes in cardiac cells could indicate lipid metabolism and cell morphology remodelling in cardiomyocytes. This led us to the hypothesis that maternal obesity caused changes in fetal heart lipid metabolism and biosynthesis. To identify the main gene pathways affected, we conducted RNA-Seq of male fetal hearts, followed by deep pathway enrichment analysis. We found 345 downregulated and 440 upregulated genes in the heart of the obese offspring ($p < 0.05$) (*Figure 5A*). Ingenuity Pathway Analysis (IPA) revealed that a set of confidently top-regulated genes ($p < 0.01$) were associated with sterol, fatty acid and carnitine metabolism (*Figure 5B*), in a scenario where PPAR-alpha and HIF1A were the main activated upstream transcriptional regulators, with signalling by *lyso*-phosphatidylcholine (LPC) abundance reduced and SREBP activity is downregulated (*Figure 5C*). Lists containing data of all regulated IPA canonical pathways and IPA upstream regulators are available in *Figure 5—source data 2* and *Figure 5—source data 3*. *Figure 5D* shows expression of genes regulated by PPAR-alpha transcriptional activity, and expression of genes mapped to the main predicted IPA pathways. We later validated the expression of key genes associated with lipid metabolism in the hearts of both male and female offspring from a completely independent cohort using qPCR (*Figure 5E*).

## The abundance of acyl-carnitines in fetal hearts is associated with maternal obesity

Having observed changes in transcriptional activity indicating increased lipid metabolism in fetal hearts in response to maternal obesity, we conducted a final experiment to investigate whether acyl-carnitine species were also affected by maternal obesity. These comprise fatty acid residues produced during beta-oxidation and are markers of mitochondrial and peroxisomal lipid metabolism. Regardless of the lack of significant differences between sex-matched obese and control offspring (*Figure 6A*), we observed increased levels of total hydroxylated acyl-carnitines in response to maternal obesity (*Figure 6B*). At the individual level, we found the hydroxylated acyl-carnitine C05-OH to be more

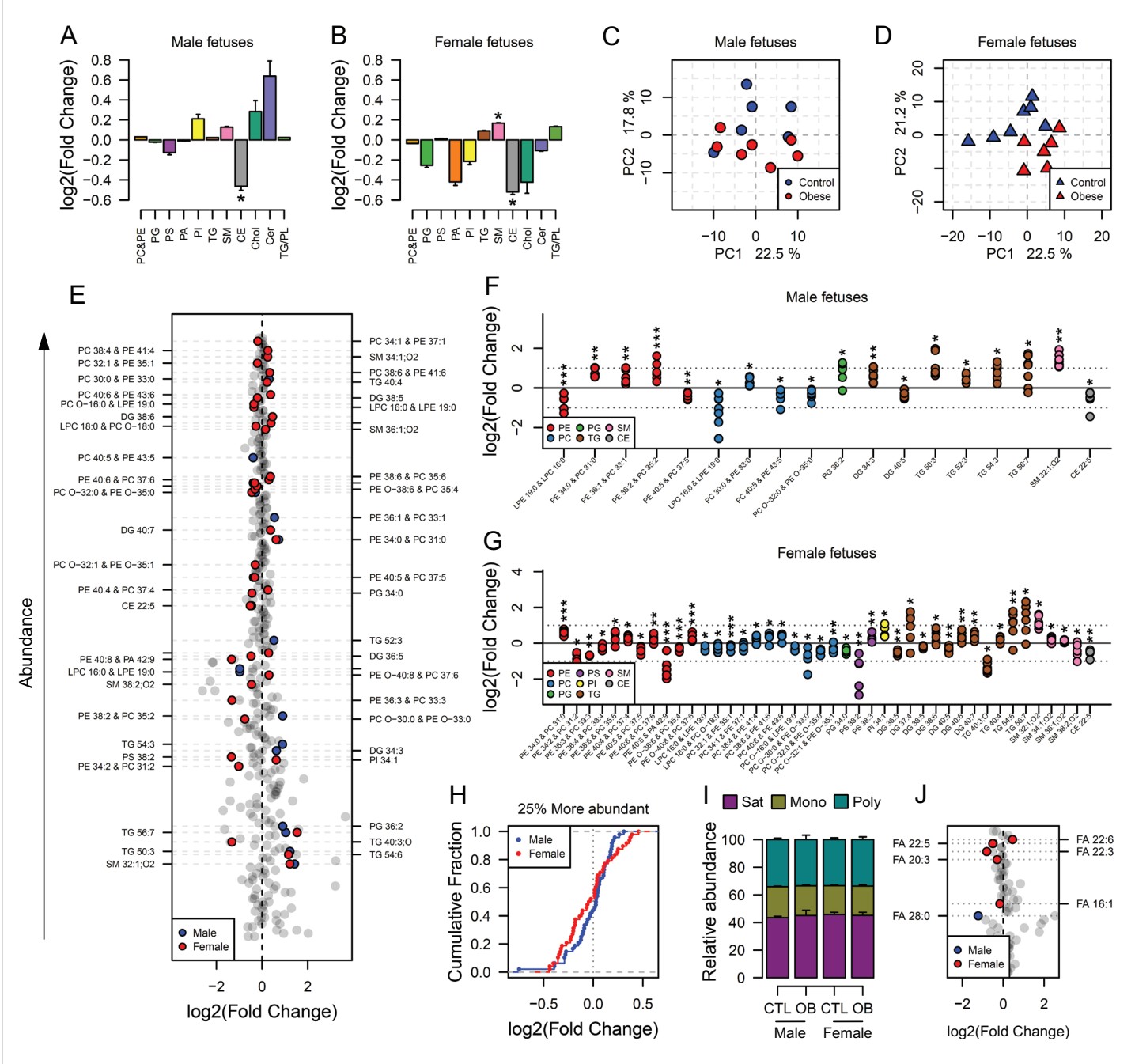

**Figure 4.** Maternal and fetal cardiac lipidome. (**A–B**) Relative changes in cardiac lipid classes in male (**A**) and female (**B**) fetuses from obese dams. Values are mean + SE. *p < 0.05 calculated by Student t-test or Mann-Whitney test. (**C–D**) PCA plots showing the PC1 and PC2 scores for individual male (**C**) and female (**D**) cardiac lipidomes. (**E**) Regulation of fetal cardiac lipid species ranked according to their abundance. Coloured dots represent statistically regulated species as calculated by univariate Student t-test (p < 0.05) and PLS-DA VIP (vip score >1) in fetal OB hearts compared to CTL. (**F–G**) Cardiac levels of regulated lipids from male (**F**) and female (**G**) fetuses of obese dams at gestational day 18.5. Each dot represents a result from one obese heart, relative to the average of results for individual lipids in the control group (straight line). Male fetuses CTL n = 6, male fetuses OB n = 7, female fetuses CTL n = 7, female fetuses OB n = 6. * p < 0.05, ** p < 0.01, *** p < 0.001 calculated by Student t-test. (**H**) Cumulative frequency of cardiac lipid species according to the log2 of the fold change in abundance between male and female fetuses from obese and control dams. (**I**) Grouped saturated, monounsaturated and polyunsaturated fatty acids content in male and female fetal hearts at gestational day 18.5. (**J**) Regulation of maternal and fetal serum fatty acids. Coloured dots represent statistically regulated fatty acids as calculated by univariate Student t-test or Mann-Whitney test (p < 0.05) in fetal OB hearts compared to CTL. Male fetuses CTL n = 8, male fetuses OB n = 6, female fetuses CTL n = 7, female fetuses OB n = 7. In figures A-B: PE, phosphatidylethanolamines/odd chain phosphatidylcholines; PC, phosphatidylcholines/odd-chain phosphatidylethanolamines; PC, phosphatidylcholines; PG, phosphatidylglycerols; PS, phosphatidylserines; PA, phosphatidic acids; PI, phosphatidylinositols; TG, monoglycerides,

*Figure 4 continued on next page*

*Figure 4 continued*

diglycerides and triglycerides; SM, sphingomyelins; CE, cholesteryl esters; Cer, ceramides; PL, phospholipids. In figures E-G, other isobaric lipids can contribute to these signals (*Supplementary file 2*). See also *Figure 4—figure supplement 1* and *Figure 4—figure supplement 2*.

The online version of this article includes the following source data and figure supplement(s) for figure 4:

**Source data 1.** Relative lipid classes abundance in fetal heart (*Figure 4A–B*).

**Source data 2.** Direct infusion high-resolution mass spectrometry of the heart (positive mode only) (*Figure 4E–H*).

**Source data 3.** Relative fatty acids classes abundance in fetal heart (*Figure 4I*).

**Source data 4.** Fatty acids abundance obtained by direct infusion high-resolution mass spectrometry of the heart (negative mode) (*Figure 4J*).

**Figure supplement 1.** Radar plots showing regulation of most abundant statistically regulated lipids in the heart of fetuses from obese pregnancies in different compartments.

**Figure supplement 2.** Scatterplots showing enrichment score (ES) and statistical significance of lipid ontology pathways from LION.

abundant in both males and females (*Figure 6C, D and E*), and C16-OH to be less abundant in females only (*Figure 6E*). C12:0 and C12:1 were also more abundant, whereas C20:0 and C22:5 were less abundant in obese female hearts (*Figure 6E*). In males, C3:0 was less abundant, and C11:0 and C15:0 were both more abundant (*Figure 6D*). In addition to the increased hydroxylated acyl-carnitine levels, a parallel experiment showed that maternal obesity caused cardiomyocytes isolated from female fetuses to oxidise more oleic acid in vitro than cardiomyocytes from control female fetuses (*Figure 6—figure supplement 1*).

## Discussion

In this study, we investigated the effect of an obesogenic in utero environment on the fetal cardiac and serum lipidome and explored if any effects were sexually dimorphic. We report for the first time unique patterns in the lipidome of the fetal heart as a consequence of maternal obesity which, despite exposure to the same in utero environment and similar serum lipid profiles, differed between male and female fetuses and occurred in the absence of sexually dimorphic changes in cardiac structure. The findings were consistent with changes in substrate availability that may affect fetal cardiac gene expression which is known to change in late gestation in preparation for birth.

Using direct infusion lipidomics, we observed unique responses to a maternal obesogenic diet between dams and offspring. This is consistent with the suggestion that the mouse placenta selectively transports lipids to the fetus, adjusting placental transfer to changes in maternal status (*Miranda et al., 2018*). In this process, maternal lipoproteins are hydrolysed at the placenta and non-esterified fatty acids are released into the fetal circulation bound to alpha-fetoproteins. Fatty acids are then taken up by the fetal liver and incorporated into lipoproteins, which are released into the fetal circulation (Reviewed by *Herrera and Desoye, 2016*). Thus, a combination of selective fatty acid transport by the placenta and differences in fetal metabolism can explain the difference between maternal and fetal serum lipid profiles in obese pregnancies.

The analysis of both male and female fetal hearts and serum is a major novelty and strength of this study, allowing for investigation of any sex-specific differences in response to maternal obesity. There is growing evidence to suggest that there are sex-specific differences in response to a suboptimal in utero environment or at least in the timing of the development of the phenotype, with the male fetus being generally more vulnerable to the long-term detrimental consequences (*Nicholas et al., 2020*; *Dearden et al., 2018*). In the current study, a greater number of significant differences between control and obese cardiac lipidomes were observed in female fetuses than were seen in males. This difference between the sexes is particularly striking, given that fetuses of both sexes are exposed to the same maternal metabolic milieu. It is not clear if these sexually dimorphic responses early in life could represent an ability of females to adapt to the environment and to protect against longer term detrimental effects. Furthermore, although several individual lipid species displayed sex-dependent regulation, a similar overall impact of maternal obesity was observed in the serum lipidomes of both male and female offspring. Therefore, there were tissue specific effects of maternal obesity on the fetal serum and cardiac lipidomes. In the serum, triglycerides were more abundant, and several phosphatidylcholines/odd chain phosphatidylethanolamines were less abundant as a consequence of maternal obesity.

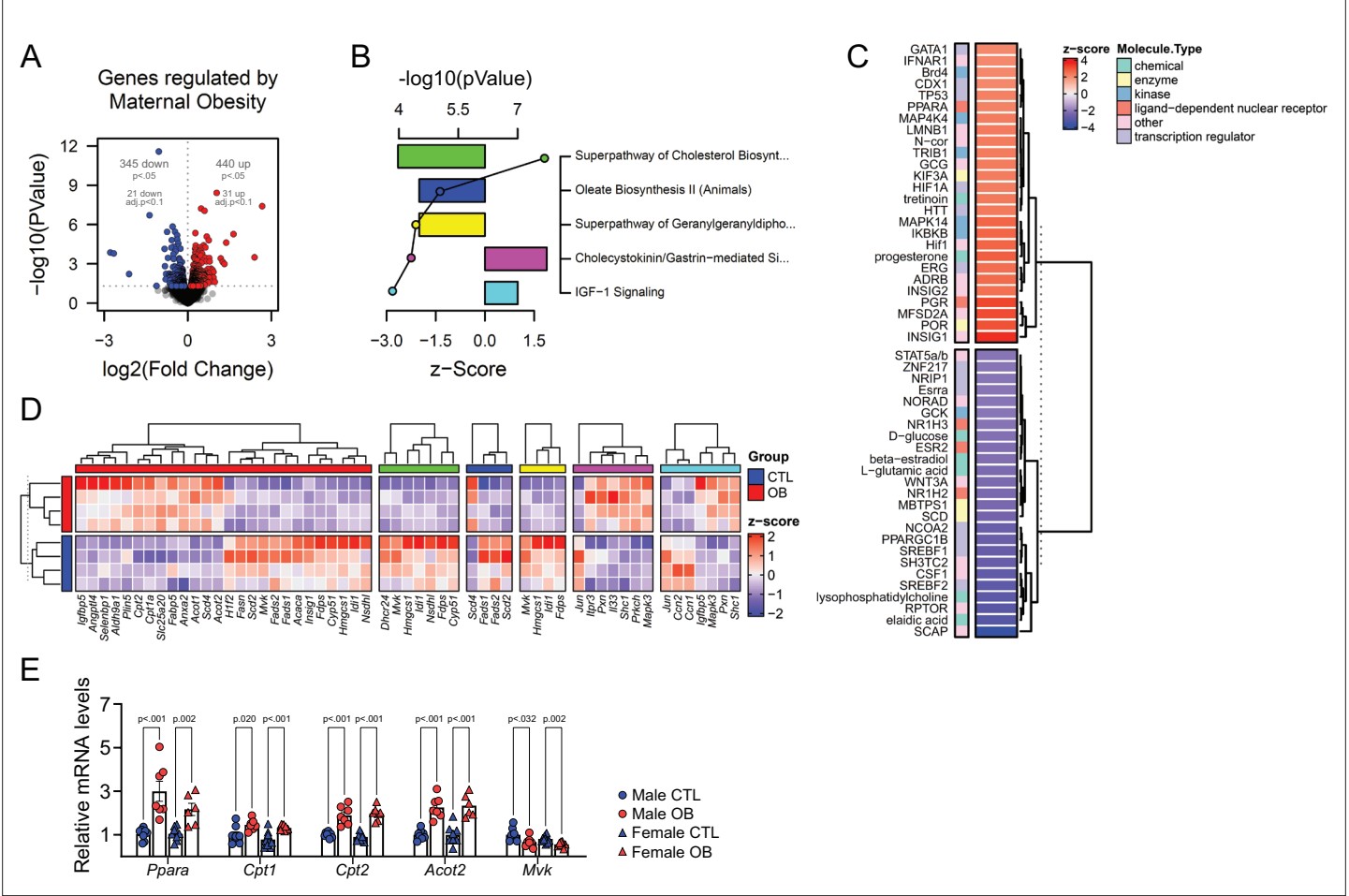

**Figure 5.** Fetal cardiac transcriptomics. (**A**) Volcano plot showing up and downregulated genes in the heart of obese fetuses. p < 0.05 and FDR < 0.1 by generalised linear models with quasi-likelihood tests. Full statistical data is available in *Figure 5—source data 1*. (**B**) Top five regulated Ingenuity Canonical Pathways predicted by analysis of cardiac transcriptome from male fetuses from obese dams compared to fetuses from control dams. A p-value cut-off of 0.01 calculated by likelihood-ratio test was used to select regulated genes included in the IPA analysis. Bars represent activation z-score per pathway; points represent p-value of enriched pathways estimated by IPA algorithm. (**C**) Activation z-score of top Ingenuity Upstream Regulators predicted by analysis of cardiac transcriptome from male fetuses from obese dams compared to fetuses from control dams. The complete lists of regulated IPA canonical pathways and IPA upstream regulators are available in *Figure 5—source data 2* and *Figure 5—source data 3*, respectively (**D**) Heatmap showing mRNA levels of genes regulated by PPAR-alpha activity (red bar), and genes mapped to 'Superpathway of Cholesterol Biosynthesis' (green bar), 'Oleate Biosynthesis II' (blue bar), 'Superpathway of Geranylgeranyldiphosphate Biosynthesis I' (yellow bar), 'Cholecystokinin/Gastrin-mediated Signalling' (pink bar) and 'IGF-1 Signalling' (light blue bar) Ingenuity Canonical Pathways in male E18.5 hearts as analysed by RNA Seq. CTL n = 4 and OB n = 4. (**E**) mRNA levels of selected markers of lipid metabolism in male and female fetal heats. Male CTL n = 8, male OB n = 8, female CTL n = 6, female OB n = 11. *p < 0.05, **p < 0.01, ***p < 0.001 by Student t-test. Diet ***p < 0.001 by factorial ANOVA. Primer sequences are available in *Table 1*.

The online version of this article includes the following source data for figure 5:

**Source data 1.** Quasi-likelihood general linear models generated from the analysis of E18.5 fetal cardiac transcriptomes (obese versus control) (*Figure 5A*).

**Source data 2.** List of IPA canonical pathways with mapped E18.5 fetal cardiac genes confidently regulated by maternal obesity (*Figure 5B*).

**Source data 3.** List of IPA upstream regulators with mapped E18.5 fetal cardiac genes confidently regulated by maternal obesity (*Figure 5C*).

**Source data 4.** Counts per million of selected genes (*Figure 5D*).

**Source data 5.** Fetal cardiac mRNA levels by qPCR (*Figure 5E*).

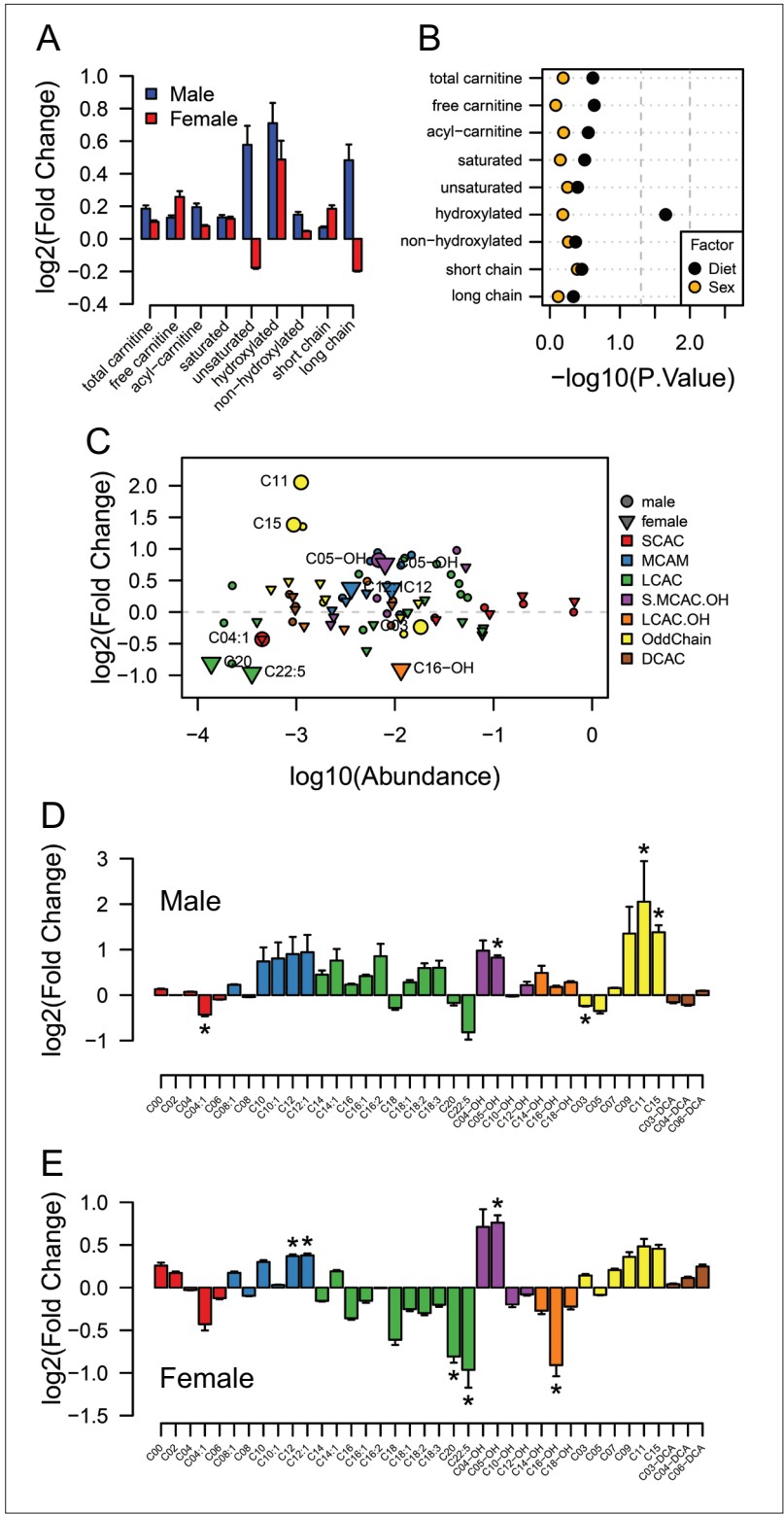

**Figure 6.** Acyl-carnitine levels in fetal hearts measured by LC-MS. (**A**) Relative changes in cardiac carnitine classes levels in male and female fetuses from obese dams. (**B**) Influence of maternal diet and sex on fetal cardiac carnitine classes levels as calculated by factorial ANOVA. (**C**) Relative fold change of individual acyl-carnitine levels in the heart of E18.5 fetuses from obese dams according to their abundance. Larger figures are acyl-carnitine species deemed as regulated with p < 0.05 by Student t-test or Mann-Whitney test. SCAC: small-chain acyl-carnitine; MCAC: medium-chain acyl-carnitine; LCAC: long-chain acyl-carnitine; S.MCAM.OH: small- and medium-chain

*Figure 6 continued on next page*

*Figure 6 continued*

hydroxy acyl-carnitine, LCAC.OH: Long-chain hydroxy acyl-carnitine; Odd Chain: acyl-carnitines with an odd chain number; DCAC: dicarboxylic acyl-carnitines. (**D–E**) Individual acyl-carnitine species levels in male (**D**) and female (**E**) fetal hearts at 18.5 days of pregnancy. See **Supplementary file 3** for list of full names. *p < 0.05 by Student t-test or Mann-Whitney test. Male fetuses CTL n = 7, male fetuses OB n = 7, female fetuses CTL n = 7, female fetuses OB n = 6.

The online version of this article includes the following source data and figure supplement(s) for figure 6:

**Source data 1.** Acyl-carnitine classes abundance in fetal heart (**Figure 6A**).

**Source data 2.** Acyl-carnitines abundance obtained by spectrometry of the heart (negative mode) (**Figure 6C–E**).

**Source data 3.** Estimated pmol [$^{14}$C]-CO$_2$/ nmol [1-$^{14}$C]-oleate produced by male and female fetal cardiomyocytes from CTL and OB dams at gestational day 18.5 (**Figure 6—figure supplement 1**).

**Figure supplement 1.** Estimated pmol [$^{14}$C]-CO$_2$/ nmol [1-$^{14}$C]-oleate produced by male and female fetal cardiomyocytes from CTL and OB dams at gestational day 18.5.

---

The observed increase in fetal serum triglyceride and decrease in phosphatidylcholine/odd chain phosphatidylethanolamines abundances is suggestive of a change in lipoprotein composition. Previous studies showed that hyperlipidaemic human serum samples have a differential increase in the concentration of lipid species, with triglycerides increasing several fold, whereas phosphatidylcholine only increased modestly (**Kuklenyik et al., 2018**). Although lipoprotein monolayers consist primarily of phosphatidylcholines enclosing a hydrophobic core containing triglycerides and cholesterol (Reviewed by **van der Veen et al., 2017**), we do not necessarily expect a positive correlation between these two lipid classes. Lipoproteins are not always spherical and thus the relationship between particle surface area and volume is not uniform. Furthermore, high-density lipoproteins (HDLs) comprise mainly phosphatidylcholines and very little triglyceride and, as the primary lipoprotein particles produced by the fetal liver are HDLs (**Herrera and Desoye, 2016**), a decrease in phosphatidylcholines without a corresponding decrease in triglyceride abundance may indicate a relative decrease in fetal HDL levels. A relative increase in triglyceride abundance in the absence of a relative increase in phosphatidylcholine may also indicate an increase in lipoprotein particle size with a smaller surface/core volume ratio in obese offspring (**Kuklenyik et al., 2018**).

Regardless of the precise mechanisms involved, the fetal serum signature indicates that maternal obesity causes deep changes to the lipids in the circulation of fetuses and thus those available to be taken up by fetal tissues. We observed that DPA – a 22:5 long-chain polyunsaturated fatty acid – from phospholipids was consistently lower in obese dams and in the serum and cardiac tissues of fetuses of both sexes. DPA is highly abundant in fish oils but can also be synthesised endogenously through eicosapentaenoic acid and arachidonic acid metabolism (**Burdge et al., 2002**). Lower serum levels of this fatty acid have recently been associated with increased markers of insulin resistance and with increased cardiovascular risk in pregnant women (**Zhu et al., 2019**). However, as far as we are aware, a direct relationship between this fatty acid and fetal health has not yet been drawn, and we believe we are the first to identify a relative reduction of phospholipid-derived DPA levels in multiple fetal compartments. Moreover, similar changes in serum and cardiac fatty acid composition of lipids might also indicate that the fetal myocardium exposed to maternal-obesity-induced stress maintains its ability to uptake circulating fatty acids.

We observed changes in the fetal cardiac transcriptome that were induced by maternal obesity, including changes in regulation of genes associated with sterol, fatty acid and carnitine metabolism, that would indicate an early shift towards fatty acid oxidation. The fetal heart relies predominantly on glycolytic metabolism, however, at birth there is a switch to lipid oxidative metabolism. Increasing oxygen levels and high lipid availability in the maternal milk both play a crucial role in activating pathways that will ultimately lead to metabolic, physiological, and morphological changes, resulting in postnatal heart maturation (**Piquereau and Ventura-Clapier, 2018**). This has been recently revisited in a study by **Sim et al., 2021**, which reported that cardiomyocyte maturation in humans is accompanied by cell specific changes in the transcriptome that are marked by the activation of oxidative metabolic pathways, such as the tricarboxylic acid cycle and respiratory electron transport chain in a sex-independent fashion.

A limitation of the current study was that we investigated the cardiac transcriptome in males only (though confirming differences in the regulation of key lipid metabolism genes also identified in the males in the females). We will therefore have missed any differences that were only present in females. Although sex differences in fetal cardiomyocytes transcriptomes exist, previous reports suggest most of the sexual dimorphism observed in transcriptional networks are identified later postnatally (*Sim et al., 2021*). In the context of nutritional programming of cardiac function, it has been established that maternal obesity causes profound changes in the adult male cardiac transcriptome, associated with activated inflammatory and fibrotic networks and decreased muscle contraction and insulin response pathways (*Ahmed et al., 2021*). However, data related to the female cardiac transcriptome was not reported in this study. These results highlight the importance of future research studying potential sexually dimorphic transcriptional changes caused by maternal obesity later in the adult heart transcriptome.

Our current observations suggest that fatty acid availability drives a premature switch in metabolism from glucose to fatty acid oxidation in the fetal heart through PPAR-alpha activation by ligands, such as docosahexaenoic acid (22:6). This indicates early metabolic, though not morphological, cardiomyocyte maturation due to a change in the availability of nutrients to fetal tissues from obese pregnancies, which could have a long-term impact on cardiac function. Changes in lipid metabolism appear to be a common consequence of exposure to an in utero obesogenic environment in different tissues, with transcriptomic analysis of livers from late gestation fetal baboons showing that maternal obesity results in altered fetal liver metabolism and lipid accumulation (*Puppala et al., 2018*).

Previous studies have suggested that maternal obesity results in placental hypoxia and a reduction in the availability of oxygen to other fetal tissues (*Wallace et al., 2019*). Consistently, we previously observed increased HIF1A protein in the obese placenta (*Fernandez-Twinn et al., 2017*), indicating lower oxygen diffusion to the fetal tissues which would be expected to impair fatty acid oxidation. Indeed, despite the increased expression of *Cpt* genes, we observed cardiac accumulation of total hydroxy acyl-carnitine, an intermediate of beta-oxidation (*Ventura et al., 1998*). CPT proteins are required for fatty acid mitochondrial import and oxidation through addition and removal of carnitine from acyl groups, allowing their transport through the intermembrane space. Accumulation of total hydroxy acyl-carnitine is also observed in diabetic hearts (*Su et al., 2005*), and may indicate impaired mitochondrial capacity to completely metabolise the surplus of fatty-acids in the matrix. Therefore, the expression of lipolytic genes alone may not be sufficient to compensate for the surplus in fatty acids and to increase the energy production and contractile potential of the fetal heart in a hypoxic environment. This may contribute to the reduced ventricular volume, as observed in the fetal hearts from obese mothers. Nevertheless, further studies are required to define the fatty acid oxidation rates and energy balance in the fetal heart.

In conclusion, we have carried out a comprehensive study of how obesity during pregnancy influences lipid availability to the fetus and consequently affects the fetal heart lipidome. From our findings, three main principles emerge: (1) There is a discrepancy between how the maternal metabolic status affects the maternal and fetal serum lipidome, an outcome likely related both to the placental actions as a selective barrier and to changes in fetal metabolism; (2) Despite being exposed to the same maternal metabolic milieu, male and female fetal hearts show distinct responses to maternal obesity, mainly at the level of fatty acid residues and individual lipid isoforms. Female heart lipidomes were generally more sensitive and exhibited greater changes than males; (3) Changes in lipid supply resulting from maternal obesity might contribute to early expression of lipolytic genes in mouse hearts, possibly contributing to the previously observed changes in heart function in adult life. The precise mechanisms by which these alterations impact on long term cardiovascular health across the life course remains to be determined.

## Materials and methods

**Key resources table**

| Reagent type (species) or resource | Designation | Source or reference | Identifiers | Additional information |
|---|---|---|---|---|
| Biological sample (*Mus musculus*) | Murine cardiac ventricles | UBS, Cambridge, UK | | Excised and frozen from *Mus musculus* |

*Continued on next page*

*Continued*

| Reagent type (species) or resource | Designation | Source or reference | Identifiers | Additional information |
|---|---|---|---|---|
| Biological sample (*Mus musculus*) | Whole murine cardiac torsos | UBS, Cambridge, UK | | Fixed *Mus musculus* torsos after culling |
| Biological sample (*Mus musculus*) | Primary murine cardiomyocytes | UBS, Cambridge, UK | | Freshly isolated from *Mus musculus* |
| Antibody | Anti-Cardiac Troponin I (mouse monoclonal) | Abcam | Cat# ab8295 | IF(1:50) |
| Antibody | Anti-mouse IgG (FITC) (goat polyclonal) | Abcam | Cat# ab6785 | IF(1:1000) |
| Antibody | Anti-PCNA (mouse monoclonal) | Abcam | Cat# ab29 | IHC(1:10,000) |
| Commercial assay or kit | Pierce Primary Cardiomyocyte Isolation Kit | Thermo-Fisher | Cat# 88,281 | |
| Commercial assay or kit | HRP/DAB (AMB) detection IHC kit | Abcam | ab64264 | |
| Commercial assay or kit | miRNeasy mini kit | Qiagen | 217,004 | |
| Commercial assay or kit | TruSeq RNA Library Preparation kit v2 | Illumina | RS-122–2001 | |
| Commercial assay or kit | High-Capacity cDNA Reverse Transcription Kit | Thermo-Fisher | 4368814 | |
| Software, algorithm | Bowtie | Bowtie | RRID:SCR_005476 | V 1.2.3 |
| Software, algorithm | Ingenuity Pathways Analysis (IPA) | Qiagen | RRID:SCR_008653 | V 42012434 |
| Software, algorithm | edgeR | Bioconductor (R) | RRID:SCR_012802 | V 3.36.0 |
| Software, algorithm | RStudio | RStudio | RRID:SCR_000432 | V 1.4 |
| Software, algorithm | Prism | GraphPad | RRID:SCR_002798 | V 9.3.0 |
| Software, algorithm | QuPath | QuPath | RRID:SCR_018257 | V 0.3.0 |
| Software, algorithm | Xcalibur | Thermo | RRID:SCR_014593 | V 4.3 |
| Software, algorithm | Inkscape | Inkscape | RRID:SCR_014479 | V 1.1.1 |
| Software, algorithm | Harmony | PerkinElmer | RRID:SCR_018809 | V 5.0 |
| Software, algorithm | ImageJ | ImageJ | RRID:SCR_003070 | V 1.53 n |
| Software, algorithm | XCMS | Bioconductor (R) | RRID:SCR_015538 | V 3.16.1 |
| Software, algorithm | Peakpicker | R | *Harshfield et al., 2019* | V 2.0 |
| Other | Horse Serum, heat inactivated | Thermo-Fisher | Cat# 26050070 | |
| Other | Fetal bovine serum (FBS) for standard applications | Thermo-Fisher | Cat#26,140 | |
| Other | Paraformaldehyde Solution, 4% in PBS | Thermo-Fisher | Cat# 15670799 | |
| Other | Animal-free blocking buffer | Vector laboratories | Cat# SP-5030–250 | |
| Other | Formalin solution, neutral buffered, 10% | Merck | HT501128 | |
| Other | Haematoxylin QS Counterstain | Vector laboratories | H-3404–100 | |
| Other | Hepes | Merck | Cat# H3375 | |
| Other | DAPI solution | Merck | 10236276001 | (1 µg/mL) |
| Other | Guanidine | Merck | CAS 50-01-1 | |
| Other | Thiourea | Merck | CAS 62-56-6 | |
| Other | Dichloromethane | Merck | CAS 75-09-2 | |
| Other | Methanol | Merck | CAS 67-56-1 | |
| Other | Triethylammonium chloride | Merck | CAS 554-68-7 | |
| Other | Ammonium Acetate | Merck | CAS 631-61-8 | |

*Continued*

| Reagent type (species) or resource | Designation | Source or reference | Identifiers | Additional information |
|---|---|---|---|---|
| Other | Chloroform | Merck | 34854–2.5 L-M | |
| Other | Butyryl-d7-L-carnitine | QMX laboratory | D-7761/0.05 | |
| Other | Hexadecanoyl-L-carnitine-d3 | QMX laboratory | D-6646/0.05 | |
| Other | Acetone | Merck | A/0560/17 | |
| Other | Acetonitrile | Merck | 2856–25 | |
| Other | Power SYBR Green PCR Master Mix | Thermo-Fisher | 4367659 | |
| Other | Fatty acid-free bovine serum albumin (BSA) | Merck | A9205 | |
| Other | DMEM – low glucose | Merck | D6046 | |
| Other | L-Carnitine | Merck | CAS 541-15-1 | |
| Other | Oleic Acid, [1–14 C]-, 50 µCi | PerkinElmer | NEC317050UC | |

## Lead contact

Further information and requests for resources and reagents should be directed to and will be fulfilled by the Lead Contact, Lucas Carminatti Pantaleão (lp435@medschl.cam.ac.uk).

## Data and code availability

The transcriptomics and lipidomics datasets generated during this study are available at GEO [GSE162185] and as supplemental material (*Figure 2—source data 3*, *Figure 3—source data 2*, *Figure 4—source data 2*, *Figure 4—source data 4* and *Figure 6—source data 2*), respectively.

## Animal handling

This research was regulated under the Animals (Scientific Procedures) Act 1986 Amendment Regulations 2012 following ethical review by the University of Cambridge Animal Welfare and Ethical Review Body (AWERB). Female C57BL/6 J mice were randomly allocated to receive either commercial standard (RM1) or a high fat diet [20% lipids (Special Dietary Services)] plus sweetened condensed milk [55% simple carbohydrates/8% lipids (Nestle, UK)] from weaning. After around 6 weeks on their respective diets, mice were mated with male counterparts and went through a first pregnancy and lactation to ensure their breeding competence. Mice were mated a second time, and first day of pregnancy was marked by the detection of a vaginal plug. On day 18.5 of gestation, pregnant mice were culled by rising $CO_2$ concentrations or cervical dislocation. Fetuses were surgically removed and immediately euthanised by laying in cold buffer. Fetal heart ventricles (comprising both left and right sides) were dissected, weighed, snap frozen and stored at −80 °C or whole fetal torsos were fixed in 10% neutral buffered formalin. Maternal blood was collected by cardiac puncture and fetal trunk blood was collected following *post-mortem* head removal. All blood samples were processed for serum separation following standard protocol.

## Cell size assessment

Fetal hearts were excised, immediately transferred to a new clean tube containing ice cold HBSS (no $Mg^{2+}$ no $Ca^{2+}$), minced into 1 mm pieces and washed twice. An enzyme cocktail from Pierce Primary Cardiomyocyte Isolation Kit containing thermolysin and papain (Thermo-Fisher, Waltham, MA, USA) was added to the minced tissue and incubated for 30 min at 37 °C. Cells were dispersed into dispersion buffer [10% horse serum [(heat inactivated) Thermo-Fisher], 5% FBS (Thermo-Fisher), 10 mM Hepes (Sigma-Aldrich, Dorset, UK) in HBSS] and strained using 70 µm strainer. An equal volume of 4% PFA (Thermo-Fisher) solution in PBS was added to the dispersion buffer (final PFA concentration: 2%) and incubated for 18 min at room temperature. Fixed cells were then centrifuged for 2 min at 700 g and the fixative was removed. Cells were washed twice with cold PBS and stored at 4 °C. Fifty thousand cells per extract were incubated with animal-free blocking buffer (Vector Laboratories, Burlingame, CA, USA) for 45 min, and with fresh blocking buffer containing 1:50 mAb to Cardiac Troponin I [ab8295 (Abcam, Cambridge, UK)] and 1:1000 pAb to Ms IgG (FITC) [ab6785 (Abcam)] for 1 hr each.

Later, cells were also incubated with DAPI solution, resuspended in fresh deionised water, plated onto poly-L-lysine coated wells of a 96-well plate and observed in Opera Phenix High Content Screening System (PerkinElmer, Waltham, MA, USA).

## Immunohistochemistry

E18.5 male and female whole fetal torsos were immersion fixed in 10% neutral formalin buffer. Fetal hearts were dissected from the fixed torsos, embedded into paraffin and sectioned in the transverse plane at 3 μm using a microtome (Leica Microsystems, Wetzlar, Germany).

Slides containing mid-cardiac sections were deparaffinised, subjected to target/antigen retrieval for 20 min at 90 °C in citrate buffer (pH 6), cooled to room temperature (20 min), rinsed with wash buffer and blocked for 10 min [(ab64264), Abcam]. Sections were then incubated with 1:10,000 mAb to PCNA [(ab29), Abcam] for 1 hr. HRP/DAB detection was then performed using a mouse specific HRP/DAB (AMB) detection IHC kit [(ab64264), Abcam] according to the manufacturer's instructions. Sections were counterstained with Haematoxylin QS (Vector), washed and mounted using a synthetic mountant. Sections were imaged using a Slide Scanner Axioscan Z1 (Zeiss, Oberkochen, Germany). DAB positive nuclei staining analyses were performed blinded using QuPath v0.2.0 (*Bankhead et al., 2017*).

## Cardiac stereology

E18.5 male and female fetal torsos were immersion fixed in 10% neutral formalin buffer. Fetal hearts were dissected from the fixed torsos, embedded into paraffin and exhaustively serial sectioned in the transverse plane at 5 μm using a microtome (Leica Microsystems, Wetzlar, Germany). 10 evenly spaced sections per heart were stained with haematoxylin and eosin, mounted using a synthetic mountant and imaged using a Slide Scanner Axioscan Z1 (Zeiss, Oberkochen, Germany). Stereological analysis was performed using ImageJ v1.53. In brief, a points grid was superimposed on images with a points area of 9000 μm$^2$ and the number of points landing on each cardiac region counted by manual analysis (*Figure 1—figure supplement 1*). The total points landing on each region was used to calculate cardiac areas for each section and, using the known distance between sections, Cavalieri's principle was applied to calculate cardiac volumes.

## qPCR and RNA sequencing

Cardiac RNA from male fetuses was extracted using a Qiazol/miRNeasy mini kit protocol (Qiagen, Hilden, Germany). Library preparation for mRNA sequencing followed manufacturer protocol of TruSeq RNA Library Preparation kit (Illumina, Cambridge, UK). Libraries were sequenced using a HiSeq 4,000 platform and raw reads were mapped to mouse genome through bowtie v1.2.3. For RT-qPCR, RNA was reverse-transcribed using a High-Capacity cDNA Reverse Transcription Kit (Thermo-Fisher). QPCR reactions were prepared using SYBR Green Master Mix (Thermo-Fisher) and specific primers. Fold changes were calculated by the 2−ΔΔCT method using *Hprt* and *Sdha* as housekeeping genes. Primers and oligonucleotides used in this study are listed in *Table 1*.

## Untargeted lipidomics – preparation of samples

Solvents were purchased from Sigma-Aldrich Ltd (Dorset, UK) of at least HPLC grade and were not purified further. Lipid standards were purchased from Avanti Polar lipids (Alabaster, AL; via Instruchemie, Delfzijl, NL) and C/D/N/ isotopes (Quebec, Canada; via Qmx Laboratories, Thaxted, UK) and used

**Table 1.** Sequence-specific primers for qPCR.

| Gene | Forward primer (5' to 3') | Reverse primer (5' to 3') |
|------|---------------------------|---------------------------|
| *Acot2* | GCCACCCCGAGGTAAAAGGA | CCACGACATCCAAGAGACCAT |
| *Cpt1* | TCCGCTCGCTCATTCCGC | TGCCATTCTTGAATCGGATGAACT |
| *Cpt2* | TCGTACCCACCATGCACTAC | GTTTAGGGATAGGCAGCCTGG |
| *Mvk* | CGGGGCAGAAGTCTCAGAAG | TTCTCAAGTTCAAGGCCGCT |
| *Ppara* | TGCAGCCTCAGCCAAGTTGAA | CCCGAACTTGACCAGCCACA |

without purification. Consumables were purchased from Sarstedt AG & Co (Leicester, UK) or Wolf Labs (Wolverhampton, UK).

The methods for preparing samples and extracting lipids used was described recently (*Furse et al., 2020*). Briefly, frozen whole fetal hearts were homogenised in a stock solution of guanidine and thiourea (6 M/1·5 M; 20× *w/v*) using mechanical agitation. The dispersions were freeze-thawed once before being centrifuged (12,000× *g*, 10 min). The supernatant was collected and frozen (–80 °C) immediately. The thawed dispersion (60 µL) and serum aliquots (20 µL) were injected into wells (96 w plate, Esslab Plate+, 2·4 mL/well, glass-coated) followed by internal standards (150 µL, mixture of Internal Standards in methanol), water (500 µL) and DMT (500 µL, dichloromethane, methanol and triethylammonium chloride, 3:1:0·005). The mixture was agitated (96-channel pipette) before being centrifuged (3·2× g, 2 min). A portion of the organic solution (20 µL) was transferred to an analytical plate (96 w, glass coated, Esslab Plate+) before being dried under Nitrogen gas. The dried films were re-dissolved (TBME, 30 µL/well) and diluted with a stock mixture of alcohols and ammonium acetate (100 µL/well; propan-2-ol: methanol, 2:1; CH3COO.NH4, 7·5 mM). The analytical plate was heat-sealed and run immediately.

## Untargeted lipidomics – direct infusion mass spectrometry

Samples were directly infused into an Exactive Orbitrap (Thermo, Hemel Hampstead, UK), using a TriVersa NanoMate (Advion, Ithaca, US) for Direct Infusion Mass Spectrometry (*Harshfield et al., 2019*; *Furse et al., 2020*). A three-part analytical method was used (*Furse et al., 2020*; *Furse and Koulman, 2019*) in which samples were ionised in positive, negative, and then negative-with-collision-induced-ionisation modes. The NanoMate infusion mandrel was used to pierce the seal of each well before an aliquot of the solution (15 µL) was collected with an air gap (1·5 µL). The tip was pressed against a fresh nozzle and the sample was dispensed using 0·2 psi (N$_{2 (g)}$). Ionisation was achieved at a 1·2 kV. The Exactive was set to start acquiring data 20 s after sample aspiration began. The data were collected with a scan rate of 1 Hz (resulting in a mass resolution of 65,000 full width at half-maximum (fwhm) at 400 *m/z*). After 72 s of acquisition in positive mode the NanoMate and the Exactive switched to negative mode, decreasing the voltage to −1·5 kV. The spray was maintained for another 66 s, after which Collision-Induced Dissociation (CID) commenced, with a mass window of 50–1000 Da, and was stopped after another 66 s. The analysis was then stopped, and the tip discarded before the analysis of the next sample began. The sample plate was kept at 10 °C throughout the data acquisition. Samples were run in row order.

Raw high-resolution mass-spectrometry data were processed using XCMS (https://www.bioconductor.org) and Peakpicker v 2.0 (an in-house R script, *Harshfield et al., 2019*). Theoretical lists of known species (by m/z) were used for both positive ion and negative ion mode (~8·5 k species including different adducts and fragmentations). Variables whose mass deviated by more than nine ppm from the expected value, had a signal-to-noise ratio of <3 and had signals for fewer than 20% of samples were discarded. The correlation of signal intensity to concentration of lipid variables found in pooled mouse heart homogenate, pooled mouse liver homogenate, and pooled human serum samples (0·25, 0·5, 1·0×) was used to identify the lipid signals the strength of which was linearly proportional to abundance (*r* > 0·75) in samples.

For the detection of fatty acids of phospholipids only, a deviations threshold of 12·5ppm was used for processing of the negative mode with CID, on a list of fatty acids of chain length 14–36 with up to six double bonds and/or one hydroxyl group. All signals greater than noise were carried forward. In this method the lipidome is not separated chromatographically but measured only by mass-to-charge ratio and therefore cannot distinguish lipids that are isobaric (identical molecular mass) in a given ionisation mode. In this study, the identification of the lipids was based on their accurate mass in positive ionisation mode according to Lipid Maps structure database (*Sud et al., 2007*). In case of multiple isobars per lipid signal, the likely identification was predicted according to the biological likelihood, full list of possible annotations can be found in the supplementary information (*Supplementary files 1 and 2*). Signals consistent with fatty acids were found in 3/3 samples checked manually. Relative abundance was calculated by dividing each signal by the sum of signals for that sample, expressed per mille (‰). Zero values were interpreted as not measured and for the remaining non-assigned values, we used a known single component projection based on nonlinear iterative partial least squares algorithm (*Nelson et al., 1996*) to impute values and populate the

dataset. Data was normalised using quantile Cyclic Loess method and statistical calculations were done on these finalised values.

## Acyl-carnitine analysis – preparation of samples

All solvents and additives were of HPLC grade or higher and purchased from Sigma Aldrich unless otherwise stated.

The protein precipitation liquid extraction protocol was as follows: the tissue samples were weighed (between 1.4–11.0 mg) and transferred into a 2 mL screw cap Eppendorf plastic tubes (Eppendorf, Stevenage, UK) along with a single 5 mm stainless steel ball bearing. Immediately, 400 µL of chloroform and methanol (2:1, respectively) solution was added to each sample, followed by thorough mixing. The samples were then homogenised in the chloroform and methanol (2:1, respectively) using a Bioprep 24–1004 homogenizer (Allsheng, Hangzhou City, China) run at speed; 4.5 m/s, time; 30 seconds for two cycles. Then, 400 µL of chloroform, 100 µL of methanol, and 100 µL of the stable isotope labelled acyl-carnitine internal standard; containing butyryl-d7-L-carnitine (order number: D-7761, QMX Laboratories Ltd. (QMX Laboratories Ltd, Essex, United Kingdom)) and hexadecanoylLcarnitine-d3 (order number: D-6646, QMX Laboratories Ltd.) at 5 µM in methanol was added to each sample. The samples were homogenised again using a Bioprep 24–1004 homogenizer run at speed; 4.5 m/s, time; 30 seconds for two cycles. To ensure fibrous material was diminished, the samples were sonicated for 30 minutes in a water bath sonicator at room temperature (Advantage-Lab, Menen, Belgium). Then, 400 µL of acetone was added to each sample. The samples were thoroughly mixed and centrifuged for 10 minutes at ~20,000 g to pellet any insoluble material at the bottom of the vial. The single-layer supernatant was pipetted into separate 2-mL screw cap amber-glass autosampler vials (Agilent Technologies, Cheadle, United Kingdom); being careful not to break up the solid pellet at the bottom of the tube. The organic extracts (chloroform, methanol, acetone composition; ~ 7:3:4, ~ 1.4 mL) were dried down to dryness using a Concentrator Plus system (Eppendorf, Stevenage, UK) run for 60 min at 60°C. The samples were reconstituted in 100 µL of water and acetonitrile (95:5, respectively) then thoroughly mixed. The reconstituted sample was transferred into a 250 µL low-volume vial insert inside a 2-mL amber glass auto-sample vial ready for liquid chromatography with mass spectrometry detection (LC-MS) analysis.

## Acyl-carnitine analysis – liquid chromatography mass spectrometry

Full chromatographic separation of acyl-carnitines was achieved using Shimadzu HPLC System (Shimadzu UK Ltd., Milton Keynes, United Kingdom) with the injection of 10 µL onto a Hichrom ACE Excel 2 C18-PFP column (Hichrom Ltd., Berkshire, United Kingdom); 2.0 µm, I.D. 2.1 mm X 150 mm, maintained at 55 °C. Mobile phase A was water with 0.1% formic acid. Mobile phase B was acetonitrile with 0.1% formic acid. The flow was maintained at 500 µL/min through the following gradient: 0 minutes_5% mobile phase B, at 0.5 minutes_100% mobile phase B, at 5.5 minutes_100% mobile phase B, at 5.51 minutes_5% mobiles phase B, at 7 minutes_5% mobile phase B. The sample injection needle was washed using acetonitrile and water mix (1:1, respectively). The mass spectrometer used was the Thermo Scientific Exactive Orbitrap with a heated electrospray ionisation source (Thermo Fisher Scientific, Hemel Hempstead, UK). The mass spectrometer was calibrated immediately before sample analysis using positive and negative ionisation calibration solution (recommended by Thermo Scientific). Additionally, the heated electrospray ionisation source was optimised to ensure the best sensitivity and spray stability capillary temperature; 300°C, source heater temperature; 420°C, sheath gas flow; 40 (arbitrary), auxiliary gas flow; 15 (arbitrary), spare gas; 3 (arbitrary), source voltage; 4 kV. The mass spectrometer scan rate set at 2 Hz, giving a resolution of 50,000 (at 200 m/z) with a full-scan range of m/z 150–800 in positive mode.

Thermo Xcalibur Quan Browser data processing involved the integration of the internal standard extracted ion chromatogram (EIC) peaks at the expected retention times: butyryl-d7-L-carnitine ([M + H]+, m/z 239.19827 at 1.20 min) and hexadecanoyl-L-carnitine-d3 ([M + H]+, m/z 403.36097 at 4.20 min). The data processing also involved the integration of the targeted individual acyl-carnitine species (m/z was [M + H]+) at their expected retention time allowing for a maximum of ±0.1 min of retention time drift: any retention time drift greater than ±0.1 min resulted in the exclusion of the analyte leading to a 'Not Found' result (i.e. zero concentration). Through the Thermo Xcalibur Quan Browser software the responses of the analytes were normalised to the relevant internal standard

response (producing area ratios), these area ratios corrected the intensity for any extraction and instrument variations. The area ratios were then blank corrected where intensities less than three times the blank samples were set to a 'Not Found' result (i.e. zero concentration). The accepted area ratios were then multiplied by the concentration of the internal standard (5 µM) to give the analyte concentrations. For tissue samples, the calculated concentrations (µM) of the analytes were then divided by the amount of tissue (in mg) used in the extraction protocol to give the final results in µM per mg of tissue extracted (µM/mg).

## Fatty acid oxidation assay

A fatty acid oxidation assay was adapted from *Huynh et al., 2014*. Briefly, cardiomyocytes isolated from fetal hearts were dispersed in primary cardiomyocyte medium and pre-incubated in standard culturing conditions. After 24 hr, filter paper saturated with 1 M NaOH was placed inside sealed wells containing the primary cultures and cells were incubated in 12.5 mM HEPES, 0.3% fatty acid-free BSA, 1 mM L-carnitine, 100 µM oleic acid medium containing 0.4 µCi/ml $^{14}$C-oleate (PerkinElmer) for 3 hr at 37 °C. Hydrochloric acid was injected into wells and the solution was incubated for an extra hour at room temperature. Disintegrations per minute from $CO_2$ were measured in the filter paper to determine oleate oxidation using a TRI-CARB 5110TR Liquid Scintillation Counter system (PerkinElmer).

## Sample-size estimation

Due to the untargeted high-throughput aspect of this study and to the scarcity of available data into fetal lipidomics, the use of a power analysis accounting for changes in fetal cardiac lipids to predict sample size was challenging. The number of animals used in the present study was therefore predicted using previously obtained data on histological assessment of the ratio left ventricle:lumen in the male fetal heart, and on the extensive track record of published studies from our research group using the same maternal obesity model employed in the current study. According to an a priori unpaired t-test power calculation, an n equal or greater than five would be required to achieve significance set at $\alpha <$ 0.05, 80% power. Also, according to the resource equation, an n equal to or greater than six results in more than 10 degrees of freedom, and is therefore adequate. We then concluded that a sample size greater than six would be necessary to show any significant changes in our study.

## Biometric markers, fatty acid oxidation, and qPCR – statistical analysis

Details of statistical analysis (statistical tests used, number of animals, identification of outliers and precision measures) can be found in the figure legends and in the figure source data. Simple Student t-test was employed to identify statistically significant differences in biometric, stereological, fatty acid oxidation and qPCR analyses, comparing control and obese groups in a sex-dependent manner. Cell size statistical analysis was conducted using linear mixed-effects model followed by Tukey's *post-hoc* test. Factorial ANOVA was employed to test offspring sex and maternal status influence on individual mRNA levels.

## RNASeq – statistical analysis

Reads per Kilobase of transcript per Million mapped reads (RPKM) were produced from RNA Sequencing raw output and statistically analysed through likelihood ratio test using edgeR version 3.36.0. Core analysis in Ingenuity Pathway Analysis application (IPA – Qiagen) was used in data interpretation and pathway enrichment. A p-value cut-off of 0.01 was used to determine genes to be mapped to IPA networks.

## Untargeted and targeted lipidomics – statistical analysis

Uni- and multivariate statistical models were created using R version 3.6.3. Multiple Shapiro-Wilk tests were carried out to identify if individual variables were normally distributed. Multiple t-tests were used to identify significant regulation of individual lipid species, and multiple t-tests or Mann-Whitney tests were used to identify individual lipid classes differences between groups when individual variables were normally or non-normally distributed. A multivariate partial-least square discriminatory analysis (PLS-DA) was also employed to identify Variable Importance in the Projection (VIP) and determine individual lipids that maximise the model classification ability. For individual lipid species, variables were deemed significantly regulated and relevant when p-value

< 0.05 and vip score >1. Individual lipid classes, acyl-carnitines and fatty acids were significantly regulated when p< 0.05. Factorial ANOVA was also used to test offspring sex and maternal status influence over individual lipid classes, and pools of fatty acids and acyl-carnitines. Prior to statistical analysis, outlier samples were identified through a combination of frequency distribution analysis, lipid classes frequency investigation, PCA and hierarchical clustering analysis. Samples with lower than 66.7% of lipid signals detected or deemed as outliers in all the aforementioned analyses failed the quality control for mass spectrometry and were excluded from the datasets and from further statistical tests. Lipid ontology enrichment analysis was carried out using LION (*Molenaar et al., 2019*). Lipid traffic analysis was conducted following previously described methods (*Furse et al., 2021*).

# Additional information

## Funding

| Funder | Grant reference number | Author |
| --- | --- | --- |
| British Heart Foundation | RG/17/12/33167 | Susan E Ozanne<br>Denise S Fernandez-Twinn<br>Lucas C Pantaleão<br>Heather L Blackmore<br>Thomas Ashmore |
| Medical Research Council | MRC_MC_UU_00014/4 | Susan E Ozanne<br>Denise S Fernandez-Twinn |
| Wellcome Trust | 208363/Z/17/Z | Susan E Ozanne |
| British Heart Foundation | FS/12/64/30001 | Elena Loche |
| British Heart Foundation | FS/18/56/35177 | Isabella Inzani |
| Biotechnology and Biological Sciences Research Council | BB/M027252/1 | Samuel Furse<br>Albert Koulman<br>Benjamin Jenkins |

The funders had no role in study design, data collection and interpretation, or the decision to submit the work for publication.

## Author contributions

Lucas C Pantaleão, Formal analysis, Investigation, Visualization, Writing – original draft, Writing – review and editing, Methodology; Isabella Inzani, Formal analysis, Investigation, Methodology, Writing – original draft, Writing – review and editing; Samuel Furse, Conceptualization, Investigation, Methodology; Elena Loche, Investigation, Conceptualization; Antonia Hufnagel, Thomas Ashmore, Heather L Blackmore, Asha A M Carpenter, Investigation; Benjamin Jenkins, Investigation, Methodology; Ania Wilczynska, Martin Bushell, Formal analysis; Albert Koulman, Conceptualization, Methodology; Denise S Fernandez-Twinn, Conceptualization, Funding acquisition, Investigation, Project administration; Susan E Ozanne, Conceptualization, Funding acquisition, Project administration, Supervision

## Author ORCIDs

Lucas C Pantaleão  http://orcid.org/0000-0002-5626-8810
Isabella Inzani  http://orcid.org/0000-0003-2186-8370
Samuel Furse  http://orcid.org/0000-0003-4267-2051
Elena Loche  http://orcid.org/0000-0002-0597-4520
Antonia Hufnagel  http://orcid.org/0000-0002-7030-4419
Martin Bushell  http://orcid.org/0000-0001-9938-2691
Albert Koulman  http://orcid.org/0000-0001-9998-051X
Denise S Fernandez-Twinn  http://orcid.org/0000-0003-2610-277X
Susan E Ozanne  http://orcid.org/0000-0001-8753-5144

## Ethics

This research was regulated under the Animals (Scientific Procedures) Act 1986 Amendment Regulations 2012 following ethical review by the University of Cambridge Animal Welfare and Ethical

Review Body (AWERB). The work carried out is approved under project licences number 80/2512 and P5FDF0206.

## Decision letter and Author response

Decision letter https://doi.org/10.7554/eLife.69078.sa1
Author response https://doi.org/10.7554/eLife.69078.sa2

# Additional files

## Supplementary files

• Supplementary file 1. Isobars and main predicted classes for m/z detected in direct infusion high-resolution mass spectrometry of the serum (positive mode only). Isobar annotations were obtained from LIPID MAPS Structure Database and a mass tolerance (m/z) threshold:±0.001 was used. For multiple isobars per m/z, biological likelihood was employed to predict the likely identification. Main classes were predicted according to the likely identification.

• Supplementary file 2. Isobars and main predicted classes for m/z detected in direct infusion high-resolution mass spectrometry of the heart (positive mode only). Isobar annotations were obtained from LIPID MAPS Structure Database and a mass tolerance (m/z) threshold:±0.001 was used. For multiple isobars per m/z, biological likelihood was employed to predict the likely identification. Main classes were predicted according to the likely identification.

• Supplementary file 3. List of names for acyl-carnitines identified in E18.5 fetal hearts by LCMS.

• Transparent reporting form

## Data availability

Sequencing data have been deposited in GEO under accession code GSE162185. All lipidomics data generated or analysed during this study are included in the manuscript and supporting files. Source data files have been provided for Figures 2, 3, 4 and 6.

The following dataset was generated:

| Author(s) | Year | Dataset title | Dataset URL | Database and Identifier |
|---|---|---|---|---|
| Loche E, Ozanne SE, Wilczynska A, Pantaleao LC | 2021 | Next-Generation Sequencing Transcriptome Quantitative Analysis of Hearts of Male Fetuses From Obese Female Mice | https://www.ncbi.nlm.nih.gov/geo/query/acc.cgi?acc=GSE162185 | NCBI Gene Expression Omnibus, GSE162185 |

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
