## [Editor Report]

The manuscript by Pantaleao et al., describes the effects of maternal diet-induced obesity on lipid composition in maternal and fetal serum and the fetal heart, and in the fetal heart transcriptome. The results presented provide insight into the still poorly understood processes influencing the long-term health of the fetus. A limitation is that this is a largely descriptive study. Nonetheless, the authors provide a detailed description of the lipid composition changes in response to maternal obesity and associated with sex.

---

## [Decision Letter]

**Decision letter after peer review:**

Thank you for submitting your article "Maternal diet-induced obesity during pregnancy alters lipid supply to fetuses and changes the cardiac tissue lipidome in a sex- dependent manner" for consideration by *eLife*. Your article has been reviewed by 2 peer reviewers, and the evaluation has been overseen by a Reviewing Editor and by a Senior Editor. The reviewers have opted to remain anonymous.

The reviewers have discussed their reviews with one another, and this decision letter to help you prepare a revised submission.

Essential revisions:

1) Figure 1. The conclusion that female obese fetal hearts have smaller cardiomyocytes is based on an increased density of nuclei. Because the fetal weight does not change, is it possible that cardiomyocyte proliferation is increased? Directly measuring cardiomyocyte size and cell proliferation would address this.

2) Figure 2—figure supplement 1. The PCA plots are compelling. Plotting data of all the embryos in one PCA plot would help visualizing potential changes in lipid profiles driven by sex or maternal diet, i.e., if sex explains lipid changes, samples could then separate by sex on PC1 and diet on PC2.

3) Figure 2. Given the trends in B and C, I wonder if combining male and females together in one group would increase the statistical power and identify significant lipid changes.

4) The authors state that "Although ceramides were more abundant at the class level in obese dam serum, no ceramide isoform was statistically significantly different between groups (Figure 2E, 2F)." This is not clear, e.g., are changes at the individual level small in magnitude? Perhaps a fisher's exact test using directionality of the changes observed at the individual level could help address this discrepancy. Also, if none are significant, why is 2F indicating significance at CE 18:2 and 22:6?

5) The Figure 2 caption says: "Each dot represents a result from one obese heart". Did the authors mean serum instead of heart?

6) Line 112, 113: It is stated that the authors "explored if a subset of lipids was transported from the maternal to the fetal circulation using linear regression". Stating something along the lines that the correlation between maternal and fetal lipid abundance/composition was assessed, would be more accurate given that actual transport was not experimentally addressed.

7) Figure 3B (lines 138-140). It is not clear if these results were significantly different.

8) Lines 172-175: The authors indicate that most lipids had a greater fold change in female compared to male hearts and they reference Figure 3—figure supplement 1 and Figure 175 4—figure supplement 1. However, these figures show a similar fold changes in both males and females. This apparent discrepancy should be clarified.

9) Figure 5. More information about the overall results of the RNAseq should be provided. How broadly was gene expression altered?

10) The reason for analyzing male fetal hearts is not clear given that levels of several individual lipid species were more strongly affected in female hearts.

11) The authors confirm deregulation of key metabolic regulators in hearts. Showing corresponding changes in protein levels would help raising potential functional involvement.

12) Regarding the RNAseq analysis, it would be interesting to place the results presented within the context of recently published analyses of global gene expression in male and female hearts, and in response to obesity during pregnancy. Just to highlight a couple of relevant examples, Sim et al., (10.1161/CIRCULATIONAHA.120.051921) proposed that sex-specific metabolic programs and cardiac maturation are influenced by progesterone-regulated pathways. Ahmed et al., (10.1016/j.molmet.2020.101116) proposed that obesity during pregnancy leads to altered metabolic pathways in the heart. Analysis fetal cardiomyocyte metabolism would help establishing the functional consequences of exposure of the fetal heart to an obesogenic environment in utero.

13) The title should state that the study is on the fetal cardiac tissue lipidome and also the species and stage of development at which the fetuses are studied.

14) Line 9 would be better stated as than males.

15) In relation to the number of nuclei per unit area, did the authors observe any change in the ratio of binucleate cells.

16) Line 139 should make it clear which OB serum, maternal or fetal, is being discussed.

17) The maternal to fetal ratio of any metabolite has limitations as an index of placental transfer. Circulating fetal levels are the result of placental transfer AND fetal metabolic activity.

18) Please check all the references. Reference at line 723 is incomplete.

19) The animal handling section should state which ventricle was studied. This is important since the right and left side of the fetal heart work against very different vascular resistance.

20) it is a pity that RNA-Seq was conducted only on the male fetus. Since there are differences in the lipidome one would expect differences in the transcriptome.

21) It would be useful to present a table of other gene pathways significantly affected by maternal obesity in addition to the five selected by the authors. This reviewer understands why these five were chosen but the reader would obviously like to know if there were or were not changes in other gene pathways such as glucocorticoid steroidogenesis, for example, in light of the fact that the authors state that cardiac cell maturation may have been accelerated. These studies are very difficult to conduct so the maximum available relevant information to the model should be provided. I suggest that a table of all the significantly affected gene pathways be included. Providing this information would give the paper more reference value for other investigators in relation to fetal programming by maternal obesity. For example knowing whether there were or were not changes in mitochondrial function and oxidative phosphorylation is very important in relation to maternal obesity and the development of the heart. Another example is the proteasome which affects cardiac function and longevity. Changes in the liver transcriptome have been published in the fetuses of obese baboons at a very equivalent stage of gestation In this paper all significantly affected pathways were listed PMID: 29516496. A cross reference would be useful.

---

## [Author Response]

Essential revisions:1) Figure 1. The conclusion that female obese fetal hearts have smaller cardiomyocytes is based on an increased density of nuclei. Because the fetal weight does not change, is it possible that cardiomyocyte proliferation is increased? Directly measuring cardiomyocyte size and cell proliferation would address this.

We thank the reviewer for these comments. We have addressed all of them with new data. We agree that the observation of implied smaller cardiomyocytes was based on results from an indirect assessment so in the revised version of the manuscript we have added new data generated by direct assessment of cell size. We collected fresh fetal (E18.5) hearts and isolated and stained cardiomyocytes using a cardiomyocyte marker and DAPI (nuclear marker). We then used High Content Microscopy (HCM) to quantify individual cell area and number of nuclei per cell (methods section lines 422-439). We found no differences in cell size between groups (Figure 1I of the revised manuscript). A representative image of the HCM analysis is also included in Figure 1—figure supplement 2. To assess potential changes in cell proliferation on gestational day 18.5, we analysed nuclear PCNA levels in mid-cardiac sections using immunohistochemistry (methods section lines 441-453). We observed no changes in nuclear PCNA levels in the heart of obese offspring (Figure 1J). To further explore if maternal obesity causes morphological changes in the fetal heart, we have performed stereological analysis of the cardiac ventricular wall volumes of 18.5-day fetuses (methods section lines 455-466). We found a significant reduction in the absolute left ventricle wall volume in both males and females (Figure 1E), of a similar magnitude to the reduction in body weight, hence left ventricle wall volumes relative to body weight were unchanged (Figure 1G, 1H).

2) Figure 2—figure supplement 1. The PCA plots are compelling. Plotting data of all the embryos in one PCA plot would help visualizing potential changes in lipid profiles driven by sex or maternal diet, i.e., if sex explains lipid changes, samples could then separate by sex on PC1 and diet on PC2.

Thank you for this very insightful comment. As suggested by the referee we have generated a unified PCA plot with the orthogonal linear transformation of both male and female data. No differences in male and female serum were identified. In both components PC1 and PC2, the separation between fetuses from control and obese dams is clearer (Figure 2—figure supplement 1C).

3) Figure 2. Given the trends in B and C, I wonder if combining male and females together in one group would increase the statistical power and identify significant lipid changes.

Thank you for your comment. We have now run statistical analysis using the merged male and female data (new Figure 2B), which increased the statistical power in the analysis of lipid classes. This reveals upregulation of the PG and PA lipid classes in the serum of fetuses from obese dams (Figure 2B).

4) The authors state that "Although ceramides were more abundant at the class level in obese dam serum, no ceramide isoform was statistically significantly different between groups (Figure 2E, 2F)." This is not clear, e.g., are changes at the individual level small in magnitude? Perhaps a fisher's exact test using directionality of the changes observed at the individual level could help address this discrepancy. Also, if none are significant, why is 2F indicating significance at CE 18:2 and 22:6?

We thank the reviewer for the comment. We apologise for any confusion that may have arisen from the abbreviations used. Our figures use the abbreviation CE to refer to cholesteryl esters (not ceramides). We have now defined CE and Cer in the figure legends for increased clarity. We observed individual changes in cholesterol esters, as shown in Figure 2 and mentioned in the text (lines 117-119), including CE 18:2 and 22:6. In contrast we do not see any differences in individual ceramide isoform levels in the maternal serum (as shown in Figure 2D in the manuscript), abbreviated to Cer in figures.

Following the reviewer’s suggestion, we have performed a Fisher's exact test to address directionality of regulation of individual ceramides in the blood of obese dams. This did not alter our original conclusion with no significant differences in individual ceramides observed (see Author response table 1).

**Author response table 1. sa2table1:** 

Lipid	Direction	CTL	OB	P.value
Cer 34:1;O2_[M + H-H2O]+	Low	5	2	0.6083916
	High	4	4	
Cer 42:1;O2_[M + H-H2O]+	Low	5	2	0.6083916
	High	4	4	
Cer 42:2;O2_[M + H-H2O]+	Low	4	3	1

5) The Figure 2 caption says: "Each dot represents a result from one obese heart". Did the authors mean serum instead of heart?

We apologise for this typographical error which has now been corrected.

6) Line 112, 113: It is stated that the authors "explored if a subset of lipids was transported from the maternal to the fetal circulation using linear regression". Stating something along the lines that the correlation between maternal and fetal lipid abundance/composition was assessed, would be more accurate given that actual transport was not experimentally addressed.

Thank you for your comment, the text has been reworded as recommended (lines 122-124*7)*

7) Figure 3B (lines 138-140). It is not clear if these results were significantly different.

Thank you for your comment. The data referred to on these lines was a general observation of the data rather than the results of a specific statistical test. The text has been updated to clarify this (lines 153156).

8) Lines 172-175: The authors indicate that most lipids had a greater fold change in female compared to male hearts and they reference Figure 3—figure supplement 1 and Figure 175 4—figure supplement 1. However, these figures show a similar fold changes in both males and females. This apparent discrepancy should be clarified.

We thank the reviewer for the comment and are happy to clarify. The previous figures showed the regulation of the lipids in the cardiac tissue of fetuses from obese pregnancies using a logarithmic scale, which made it difficult to visualise the sex differences. We’ve now transformed this data back to a linear scale to make the differences more visible. A further point to note is that the lipids shown in figure 3 - figure supplement 1 were the most abundant esterified lipids identified by direct infusion mass spectrometry in the fetal hearts. Therefore, even small changes in their expression represent a large change in the cardiac lipid balance. The revised presentation of the figures now hopefully highlights that in females, 9 out of the 16 species selected were more strongly downregulated (PC O−32:0 and PE O−35:0, PC 32:1 and PE 35:1, PC 34:1 and PE 37:1 and PE 36:4 and PC 33:4) or upregulated (PC 38:4 and PE 41:4, PC 38:6 and PE 41:6, PC 40:6 and PE 43:6, PE 38:6 and PC 35:6 and SM 34:1;O2); and that the regulation of one lipid species (PC 30:0 and PE 33:0) was stronger in males.

9) Figure 5. More information about the overall results of the RNAseq should be provided. How broadly was gene expression altered?

We thank the reviewer for the comment and are happy to expand on the data from the RNAseq presented in the manuscript. We have added a volcano plot to Figure 5A which graphically shows that 785 genes were regulated in the hearts of the obese offspring with a p<0.05. We have also updated the Results section and included a table containing statistical analysis data, based on multiple generalized linear models with quasi-likelihood tests (Figure 5–source data 1).

10) The reason for analyzing male fetal hearts is not clear given that levels of several individual lipid species were more strongly affected in female hearts.

We thank the reviewer for their comment, and we agree that RNAseq data from the female hearts would be informative. Unfortunately, the RNASeq analysis was performed in parallel with the lipidomic analysis, so we did not know the results of the lipidomic analysis when we carried out the RNAseq. We only had capacity to run the RNAseq analysis on one sex with validations carried out in both sexes. We made the decision to use male hearts based on previous literature which suggests greater programming effects are observed in male offspring compared to females. We have indicated in the discussion that this is a limitation of the current study (lines 324-327).

11) The authors confirm deregulation of key metabolic regulators in hearts. Showing corresponding changes in protein levels would help raising potential functional involvement.

Thank you for your comment. We agree that identifying changes in proteins is a step closer to a functional readout (though again not perfect). Our approach to carry out a pathway analysis of RNASeq data allowed us to identify pathway deregulation as a cumulative effect of small changes across a large number of genes. It is unfortunately not possible to adopt the same approach at the protein level given the limited size of the fetal heart (a fetal mouse heart only weights approximately 5-7 mg) combined with the reduced sensitivity and semi-quantitative nature of western blotting. We attempted to carry out western blotting analysis of PPARA (one of the key metabolic regulators identified). Unfortunately, despite trialling and attempting to optimise the use of several different antibodies against PPARA from different suppliers, none were of sufficient quality to enable us to carry out any form of quantitative blotting analysis.

12) Regarding the RNAseq analysis, it would be interesting to place the results presented within the context of recently published analyses of global gene expression in male and female hearts, and in response to obesity during pregnancy. Just to highlight a couple of relevant examples, Sim et al., (10.1161/CIRCULATIONAHA.120.051921) proposed that sex-specific metabolic programs and cardiac maturation are influenced by progesterone-regulated pathways. Ahmed et al., (10.1016/j.molmet.2020.101116) proposed that obesity during pregnancy leads to altered metabolic pathways in the heart. Analysis fetal cardiomyocyte metabolism would help establishing the functional consequences of exposure of the fetal heart to an obesogenic environment in utero.

We thank the reviewer for highlighting these manuscripts which we have now included in the discussion. It is noteworthy that Sim *et al.*, suggest that expression and increased activity of sex hormone receptors mediate maturation in human cardiomyocytes, but not in mice. The authors report that androgen receptor mRNA is barely detectable, and that progesterone receptor is undetectable in mouse cardiomyocytes, and that in *Pgr* KO mice cardiomyocyte development is not affected.

We agree that the analysis of cardiomyocyte lipid metabolism would contribute to the establishment of the functional consequences of in utero exposure to maternal obesity. Indeed, we have carried out an experiment to investigate fatty acid oxidation through assessing ^14^C-oleate complete oxidation by isolated cardiomyocytes in 3 hours. This revealed that maternal obesity increased fatty acid oxidation in female cardiomyocytes but not males (Figure 6—figure supplement 1). We also analysed the levels of hydroxylated acyl-carnitine species as markers of fatty acid metabolism in cardiac tissue and this suggested increased fatty acid metabolism in hearts of fetuses from obese dams (Figure 6B of revised manuscript).

13) The title should state that the study is on the fetal cardiac tissue lipidome and also the species and stage of development at which the fetuses are studied.

We thank the reviewer for their comment. We have added the requested information to the title.

14) Line 9 would be better stated as than males.

Thank you for your feedback, the text has been amended.

15) In relation to the number of nuclei per unit area, did the authors observe any change in the ratio of binucleate cells.

We thank the reviewer for their comment. As mentioned in response to question 1, we have performed High Content Microscopy (HCM) analysis on isolated fetal cardiomyocytes stained with DAPI. Although performing an extensive visual inspection of thousands of cardiomyocytes per sample, we did not observe any notable number of binucleated cells in any group. This is an expected result as in developing mice, cardiomyocyte binucleation does not usually appear until at least postnatal day 3, and only grows in prevalence during early postnatal life.

16) Line 139 should make it clear which OB serum, maternal or fetal, is being discussed.

We apologise for the lack of clarity. The line has been updated to specify both dam and fetal serum are being discussed.

17) The maternal to fetal ratio of any metabolite has limitations as an index of placental transfer. Circulating fetal levels are the result of placental transfer AND fetal metabolic activity.

We thank the reviewer for their comment. We agree that circulating fetal lipid levels are affected by both fetal metabolism and maternoplacental transfer. The text has been updated to make this point clearer (lines 263-265).

18) Please check all the references. Reference at line 723 is incomplete.

We apologise that the stated reference was incomplete, all references have been checked for completeness.

19) The animal handling section should state which ventricle was studied. This is important since the right and left side of the fetal heart work against very different vascular resistance.

We thank the reviewer for this comment. Due to the small size of the mouse fetal hearts the left and right ventricles were not dissected separately and both sides were included in molecular analyses. The text has been updated to clarify this (lines 400-402).

20) it is a pity that RNA-Seq was conducted only on the male fetus. Since there are differences in the lipidome one would expect differences in the transcriptome.

We thank the reviewer for this comment which we have responded to in detail under point 10.

21) It would be useful to present a table of other gene pathways significantly affected by maternal obesity in addition to the five selected by the authors. This reviewer understands why these five were chosen but the reader would obviously like to know if there were or were not changes in other gene pathways such as glucocorticoid steroidogenesis, for example, in light of the fact that the authors state that cardiac cell maturation may have been accelerated. These studies are very difficult to conduct so the maximum available relevant information to the model should be provided. I suggest that a table of all the significantly affected gene pathways be included. Providing this information would give the paper more reference value for other investigators in relation to fetal programming by maternal obesity. For example knowing whether there were or were not changes in mitochondrial function and oxidative phosphorylation is very important in relation to maternal obesity and the development of the heart. Another example is the proteasome which affects cardiac function and longevity. Changes in the liver transcriptome have been published in the fetuses of obese baboons at a very equivalent stage of gestation In this paper all significantly affected pathways were listed PMID: 29516496. A cross reference would be useful.

We thank the reviewer for the comment and are in complete agreement that it is important to share the data as widely as possible [hence why we have deposited the RNAseq data in GEO (GSE162185)]. We would like to clarify that the pathways shown in Figure 5B are the top 5 pathways regulated by maternal obesity in the fetal heart, according to IPA’s algorithm (p<0.001), hence why we decided to focus on lipid metabolism when discussing the results from our transcriptomics data. Following the reviewer’s suggestion and for greater clarity, we have now included two supplementary tables (Figure 5–source data 2 and Figure 5–source data 3) listing all canonical pathways and upstream regulators recorded in the IPA analysis, including those significantly altered.

Regarding glucocorticoid signalling and steroidogenesis pathways, our IPA analysis tested but did not detect significant changes (based on a threshold of p < 0.05) in either “Glucocorticoid Receptor Signalling” or HSD11B1 activity.

Regarding mitochondrial oxidation, the IPA algorithm listed “Mitochondrial dysfunction” as a significantly affected pathway (p=0.040), but it failed to provide us with a z-score for this analysis. This might indicate that there are not enough changes in the expression of genes mapped to this pathway, or that the expected direction of regulation for each gene is not accurate for the algorithm to confidently predict activation directionality.

Regarding changes in the proteasome activity, there were no significant changes in related pathways.

Thank you for the suggested reference, which has been incorporated into our discussion of the transcriptomics data (lines 341-344).